# eDNA-stimulated cell dispersion from *Caulobacter crescentus* biofilms upon oxygen limitation is dependent on a toxin–antitoxin system

Cecile Berne, Sébastien Zappa, Yves V Brun*

Département de microbiologie, infectiologie et immunologie, Université de Montréal, Montréal, Canada

**Abstract** In their natural environment, most bacteria preferentially live as complex surface-attached multicellular colonies called biofilms. Biofilms begin with a few cells adhering to a surface, where they multiply to form a mature colony. When conditions deteriorate, cells can leave the biofilm. This dispersion is thought to be an important process that modifies the overall biofilm architecture and that promotes colonization of new environments. In *Caulobacter crescentus* biofilms, extracellular DNA (eDNA) is released upon cell death and prevents newborn cells from joining the established biofilm. Thus, eDNA promotes the dispersal of newborn cells and the subsequent colonization of new environments. These observations suggest that eDNA is a cue for sensing detrimental environmental conditions in the biofilm. Here, we show that the toxin–antitoxin system (TAS) ParDE$_4$ stimulates cell death in areas of a biofilm with decreased $O_2$ availability. In conditions where $O_2$ availability is low, eDNA concentration is correlated with cell death. Cell dispersal away from biofilms is decreased when *parDE*4 is deleted, probably due to the lower local eDNA concentration. Expression of *parDE*4 is positively regulated by $O_2$ and the expression of this operon is decreased in biofilms where $O_2$ availability is low. Thus, a programmed cell death mechanism using an $O_2$-regulated TAS stimulates dispersal away from areas of a biofilm with decreased $O_2$ availability and favors colonization of a new, more hospitable environment.

*For correspondence:
yves.brun@umontreal.ca

Competing interest: The authors declare that no competing interests exist.

## Editor's evaluation

In this work, the authors present compelling evidence that a toxin-antitoxin system contributes to biofilm dispersal under oxygen limited conditions. This work makes important contributions to two areas of microbial physiology; functional understanding of toxin-antitoxin systems, which have remained largely elusive, and mechanistic regulation or biofilm dispersal, is a critical, but less understood aspect of biofilm physiology.

## Introduction

Biofilms are multicellular communities attached to a surface, where complex exchanges and interactions occur between the different members. Biofilms first start with the attachment of individual bacterial cells to a surface and then grow into a more complex community when attached bacteria divide and new ones join. The biofilm lifestyle is considered beneficial for bacteria, as they usually provide protection from xenobiotic stresses and predators, and increase collective nutrient availability (*Flemming et al., 2016*). However, when conditions deteriorate, cells can leave the biofilm through a process referred to as dispersal, disseminate to new environments, and form new biofilms, enabling

**eLife digest** Bacteria are more social than what had long been expected. While they can swim around on their own, most of them in fact settle down as part of a surface-bound community. The plaque on our teeth and the gooey deposit in our bathroom pipes are the visible results of this communal lifestyle. Inside this slimy 'biofilm', cells share resources and are protected from toxic substances such as antibiotics. However, being tied to one spot is not always a good thing: it may be advantageous for a cell in a biofilm to strike out on its own and resume 'single life' if local conditions deteriorate.

*Caulobacter crescentus* bacteria do not always have this choice, as adult cells in this species become permanently glued into place upon joining a biofilm. When these divide, however, their daughters have a choice: swim away, or stick with the group. Previous research has shown that this decision is influenced by the health of the community. Dying cells release DNA fragments which disable the structures allowing newborn cells to adhere to the environment, and a high mortality rate in the biofilm therefore forces unattached cells to leave the colony.

Berne et al. wanted to build on these results and examine how exactly cells die in the biofilm. In particular, the deaths could be sudden and random, with the bacteria succumbing to injury; or they could result from cells activating one of their built-in self-destruct programs. To investigate this question, genetically modified *C. crescentus* bacteria were grown in the laboratory and exposed to different environments. Combining genetic and microscopic approaches revealed that as a biofilm becomes too crowded, certain individuals self-destruct via a cell death program known as the toxin-antitoxin system. Further experiments showed that low oxygen availability was the signal that triggered self-destruction. Drops in oxygen levels can happen when the environment becomes hostile or when a colony is overpopulated. The results by Berne et al. therefore suggest that by triggering self-destruction in certain members of the community, reduced oxygen access leads to newborn cells swimming away, which in turn prevents further overcrowding and allows new, more hospitable locations to be colonized.

Biofilms are of growing interest in a wide range of human industries, but they also present formidable challenges. This is particularly the case in healthcare, as they tend to infest medical devices and help disease-causing species to resist treatments. Understanding how bacteria are encouraged to join or leave their colony is necessary to better control biofilms to our advantage.

the colonization of new niches (*Guilhen et al., 2017*). Dispersal is triggered in response to various environment and biological cues, and understanding its regulation is important to determine how biofilms can be controlled.

Many Alphaproteobacteria use a strong polar adhesin to irreversibly attach to surfaces and form biofilms (*Berne et al., 2015*; *Berne et al., 2018b*), with *Caulobacter crescentus* holdfast being the best characterized example. *C. crescentus* has a dimorphic life cycle, where each division cycle yields a sessile mother stalked cell and a motile daughter swarmer cell. Newborn motile swarmer cells bear a single flagellum and multiple pili at the new pole. After the cell cycle progresses beyond a certain point, or upon contact with a surface, the newborn cells secrete a holdfast at the same pole and differentiate into stalked cells by retracting their pili, ejecting their flagellum, and synthesizing a thin cylindrical extension of the cell envelope called the stalk, which pushes the holdfast away from the cell body. While the chemical composition of the holdfast is not entirely elucidated, it is composed of polysaccharides with four different monosaccharide constituents, as well as DNA and peptide molecules of unknown nature (*Merker and Smit, 1988*; *Hernando-Pérez et al., 2018*; *Hershey et al., 2019*). Holdfast is an extremely strong bioadhesin (*Tsang et al., 2006*; *Berne et al., 2013*) crucial for irreversible cell adhesion to solid surfaces (*Ong et al., 1990*; *Bodenmiller et al., 2004*), colonization of air–liquid interfaces (*Fiebig, 2019*), and biofilm formation (*Entcheva-Dimitrov and Spormann, 2004*).

In some bacterial species, extracellular DNA (eDNA) plays a stabilizing role in the biofilm matrix (*Okshevsky and Meyer, 2015*; *Campoccia et al., 2021*). In contrast, we previously showed that *C. crescentus* eDNA produced via cell lysis negatively regulates biofilm formation and stimulates cell dispersal. eDNA binding to unattached holdfasts inhibits their adhesiveness, thereby inhibiting cell

attachment to surfaces (*Berne et al., 2010*). In contrast, eDNA does not dislodge previously bound holdfasts. Therefore, eDNA prevents newborn swarmer cells from joining mature biofilms, but does not dissociate existing biofilms. Because inhibition by eDNA is proportional to its concentration, we proposed that eDNA serves as a rheostat-like environmental cue to trigger dispersal when conditions are detrimental and cause cell death. However, it was not known if eDNA release is the simple consequence of random cell death occurring in the biofilm as conditions worsen, or if it is the result of an active mechanism, such as programmed cell death (PCD) (*Berne et al., 2010*; *Kirkpatrick and Viollier, 2010*).

In this study, we demonstrate that cell death and eDNA release in a biofilm are regulated by a PCD mechanism that responds to oxygen availability. PCD in bacteria includes all genetically encoded mechanisms that lead to cell lysis (*Lewis, 2000*; *Bayles, 2014*). Toxin-antitoxin systems (TAS) are important regulators of PCD (*Rice and Bayles, 2008*; *Peeters and de Jonge, 2018*). These systems are comprised of a stable toxin and its unstable antitoxin cognate. The antitoxin molecule usually antagonizes the toxin under 'steady state' growth conditions; but, in PCD-triggering conditions, the antitoxin is inactivated, leading to an excess of free toxins that target key cellular processes in response to various environmental signals (*Harms et al., 2018*; *Wang et al., 2021*). There are currently eight types of TAS described in bacteria. The classification depends on the nature of the antitoxin (RNA in types I, III, and VIII, or small protein in the other TAS types), and the toxin (small protein in all but type VIII where the toxin is a small RNA), and how the antitoxin neutralizes the toxin activity (*Song and Wood, 2020a*, *Singh et al., 2021*; *Srivastava et al., 2021*).

TAS are widespread in bacterial and archaeal genomes, but despite their abundance, the biological relevance of most TAS is still elusive (*Fraikin et al., 2020*). TAS were first described as plasmid addiction modules that ensure plasmid stabilization via post-segregational killing of plasmid-free cells (*Ogura and Hiraga, 1983*; *Gerdes et al., 1986*). TAS have also been shown to promote addiction to certain chromosomally encoded elements such as integrative conjugative elements (*Wozniak and Waldor, 2009*) or CRISPR-*cas* loci (*Li et al., 2021*). In addition, TAS have been described as defense mechanism against phage infection where host translation is inhibited by the phage (*Pecota and Wood, 1996*; *Fineran et al., 2009*; *Song and Wood, 2020b*, *LeRoux and Laub, 2022*; *Vassallo et al., 2022*). In bacterial cells that lose their plasmid/chromosomal element encoding the TAS, or that are infected by phage, the amount of labile antitoxin rapidly decreases, leading to toxin activation and subsequent cell death. In addition to the well accepted role of TAS in plasmid addiction and phage exclusion, TAS have been linked to diverse physiological responses, such as biofilm formation, stress response, and persistence (*Kamruzzaman et al., 2021*), although this is still debated (*Ronneau and Helaine, 2019*; *Wade and Laub, 2019*; *Song and Wood, 2020a*, *Jurėnas et al., 2022*). Many TAS have been reported to be transcriptionally upregulated under environmental stress conditions (*Jurėnas et al., 2022*), but this increase does not necessarily trigger liberation of an active toxin (*LeRoux et al., 2020*).

Among the 18 TAS identified in the *C. crescentus* genome (*Ely, 2021*), 13 have been studied experimentally, and belong to four different groups: (1) four paralogous RelBE (*Fiebig et al., 2010*) operons and one HigBA (*Kirkpatrick et al., 2016*) operon, belonging to the type II systems where the toxins (RelE or HigB) are known to be mRNA endonucleases; (2) four type II systems belonging to the ParDE family (*Fiebig et al., 2010*), where ParE toxins are usually DNA gyrase inhibitors; (3) three paralogs of HipBA, also a type II system, where the HipA toxins inhibit protein synthesis (*Huang et al., 2020*; *Zhou et al., 2021*); and (4) SocAB, the only member of the type VI TAS described so far, where the SocB toxin directly inhibits DNA replication (*Aakre et al., 2013*). The environmental conditions that trigger any of these TAS and their biological function are not yet fully identified.

In this study, we show that the ParDE$_4$ TAS is involved in PCD and eDNA release in *C. crescentus* biofilms where it stimulates cell dispersal. We show that areas of a biofilm with decreased $O_2$ availability experience more cell death. Cell viability is improved in a Δ*parDE$_4$* mutant biofilm, especially in areas of decreased $O_2$ availability, generating less cell lysis and less eDNA release. We also show that cell dispersal is decreased when *parDE$_4$* is deleted, probably due to the lower local eDNA concentration. Expression of *parDE$_4$* is positively regulated by $O_2$ and the expression of this operon is decreased in biofilms where $O_2$ availability is low. Thus, PCD by an $O_2$-regulated TAS stimulates dispersal away from areas of a biofilm with decreased $O_2$ availability.

## Results

### The ParDE$_4$ TAS is involved in biofilm inhibition and eDNA release of *C. crescentus* grown under static conditions

We previously showed that, in *C. crescentus*, eDNA is a cue that can trigger biofilm inhibition and dispersion by binding to holdfasts and reducing their adhesiveness. This mechanism is a result of cell lysis and eDNA release in the biofilm (*Berne et al., 2010*). To investigate if this eDNA release is the product of a specific PCD mechanism, we tested if a TAS was involved in promoting cell death in the biofilm, as previously suggested (*Kirkpatrick and Viollier, 2010*). If such a TAS is inactivated, one should observe less cell death, less eDNA release, and more biofilm formation. We examined the four ParDE-like and four RelBE-like individual in-frame deletion mutants previously described (*Fiebig et al., 2010*), as well as mutants lacking the four ParDE ('All *parDE⁻* ' mutant), the four RelBE ('All *relBE⁻* ' mutant) and the eight ParDE/RelBE operons ('All *parDE⁻ All relBE⁻* ' mutant'), for their ability to form biofilms compared to *C. crescentus* CB15 wild-type (WT). For these static biofilm assays, we grew cells in two-ml plastic microfuge tubes sealed with AeraSeal breathable film, to allow for gas exchange, and incubated them statically at 30°C (*Figure 1*). We defined these growth conditions as 'moderate aeration'. All the tested mutants grew similarly to WT under these conditions (*Figure 1—figure supplement 1*).

We tested the ability of the TAS mutants to form biofilms after 48 hr, and quantified cell death and eDNA release under these growth conditions (*Figure 1*). Among single mutants, Δ*parDE$_4$* was the only strain that behaved differently compared to WT. The percentage of dead cells was lower in this mutant and it produced ~30% more biofilm than the other strains (*Figure 1A, B*). Furthermore, it released only about half of the amount of eDNA in the planktonic phase compared to WT (*Figure 1C* and *Figure 1—figure supplement 1B*). The All *parDE⁻* and the All *parDE⁻ All relBE⁻* strains, where all four *parDE* operons and all *parDE* plus all *relBE* operons were deleted respectively, behaved like the Δ*parDE$_4$* single deletion mutant (*Figure 1*). These results suggest that ParDE$_4$ plays a role in cell death and eDNA release under our experimental conditions and that the observed changes in eDNA concentration yield differences in biofilm regulation. To test if this phenotype was specific for *C. crescentus* cells that are able to form biofilms, we deleted the *hfsDAB* holdfast synthesis cluster in the Δ*parDE$_4$* background to generate a strain unable to produce holdfast, and therefore unable to adhere to surfaces and form biofilms. The double mutant Δ*parDE$_4$* Δ*hfsDAB* phenocopied the Δ*parDE$_4$* strain, with lower eDNA and lower proportion of dead cells (*Figure 1—figure supplement 2*). These results indicate that the function of ParDE$_4$ does not require cells to be adhered to a surface and suggest that it might be responding to the differences in medium aeration as described in a later section.

### The ParDE$_4$ TAS plays a role in cell death in mature biofilms of *C. crescentus*

The *parDE$_4$* operon is composed of the *parD$_4$* antitoxin gene (CC2985/CCNA_03080) and the *parE$_4$* toxin gene (CC2984/CCNA_03079), overlapping by 21 bp (*Nierman et al., 2001*; *Fiebig et al., 2010*; *Marks et al., 2010*). To assess the role of ParDE$_4$ in cell death, eDNA release, and biofilm formation over time, we monitored biofilm formation on sterile microscopy-grade clear polyvinyl chloride (PVC) strips grown under moderate aeration as depicted in *Figure 1A*. Over time, we also quantified eDNA release and cell death occurring in WT and Δ*parDE$_4$* (*Figure 2* and *Figure 2—figure supplement 1*). Cell death was reduced in Δ*parDE$_4$* biofilms compared to WT, especially at longer time points when the biofilm reached maturation (*Figure 2A, B*). In addition, less eDNA was released in these mutant cultures (*Figure 2C*). We also observed an increase in attached biomass in the Δ*parDE$_4$* mutant (*Figure 2D*). These results support our previous findings that biofilm inhibition, eDNA release, and cell death are correlated (*Berne et al., 2010*). Furthermore, these results indicate that ParDE$_4$ is involved in stimulating cell death and eDNA release, yielding a change in biofilm formation. Since eDNA stimulates dispersal from the biofilm (*Berne et al., 2010*), both the reduced cell death and eDNA release in the Δ*parDE$_4$* mutant might contribute to the increased biofilm formation.

### The ParD$_4$ antitoxin protects against cell death in the biofilm

In TAS, cell death usually occurs when there is an imbalance in steady state levels of toxins and antitoxins produced in the cell (*Harms et al., 2018*). To assess the role of ParD$_4$ antitoxin expression, we

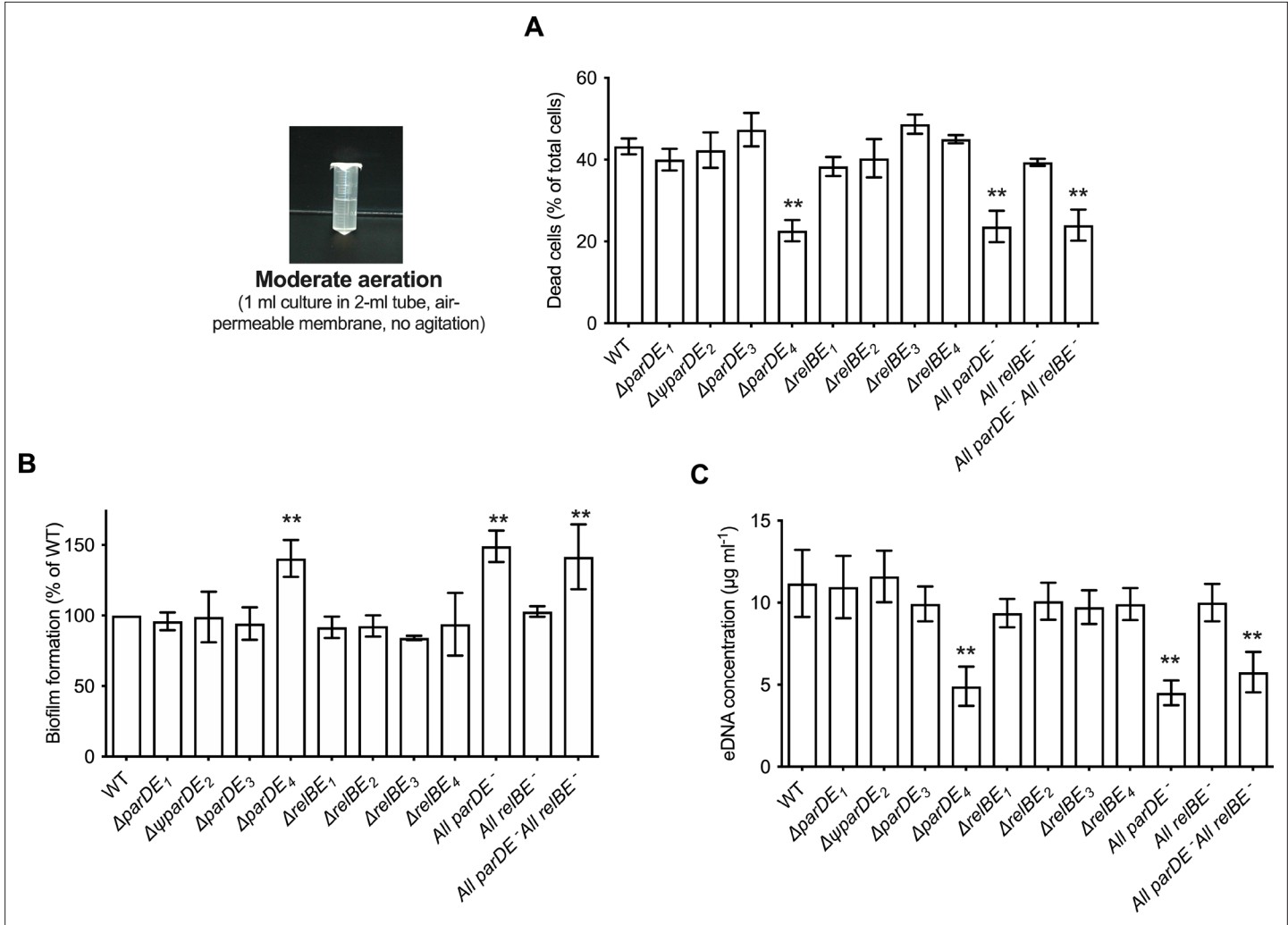

**Figure 1.** Role of the eight toxin–antitoxin systems (TAS) in cell death, extracellular DNA (eDNA) release, and biofilm formation. *C. crescentus* WT and the different TAS in-frame deletion mutants were grown for 48 hr under moderate aeration conditions at 30°C in M2G medium, as depicted on the left. (**A**) Percentage of dead cells in the planktonic phase; results are expressed as a percentage of the total cells (live + dead) in the sample, quantified using the BacLight Live/Dead kit. (**B**) Biofilm formation, quantified by crystal violet staining; results are expressed as a percentage of biofilm formed compared to WT. (**C**) Quantification of eDNA released in the planktonic phase, using PicoGreen. Results are given as the average of four independent experiments, each run in duplicate, and the error bars represent the standard error of the mean (SEM). Statistical comparisons are calculated using Student's unpaired *t*-tests; only samples statistically different from WT are shown. **p < 0.01.

The online version of this article includes the following figure supplement(s) for figure 1:

**Figure supplement 1.** ParDE$_4$ is involved in extracellular DNA (eDNA) release under moderate aeration conditions.

**Figure supplement 2.** Biofilm formation is dispensable for ParDE$_4$-mediated cell death and extracellular DNA (eDNA) release.

expressed it using the low copy xylose-inducible replicating plasmid pMT686 (*Fiebig et al., 2010*; *Thanbichler et al., 2007*) and monitored biofilm formation and eDNA release when the antitoxin is constitutively expressed. When we constitutively expressed the ParD$_4$ antitoxin in WT, biofilm formation was increased and eDNA concentration was decreased (*Figure 3A*). However, in a Δ*parE$_4$* mutant lacking the ParE$_4$ toxin, there was no effect of *parD$_4$* expression on biofilm formation and eDNA concentration (*Figure 3B*), showing that (1) ParD$_4$ has a protective effect against cell lysis and eDNA release, thereby enhancing biofilm formation, and (2) this effect depends on the presence of the toxin ParE$_4$.

Next, we wanted to determine if the behavior of the Δ*parDE$_4$* mutant was due to lack of cell death and eDNA release, or also possibly to an impaired response to eDNA biofilm inhibition in the Δ*parDE$_4$* mutant. We tested how WT and Δ*parDE$_4$* behaved in the presence of exogenous eDNA by monitoring

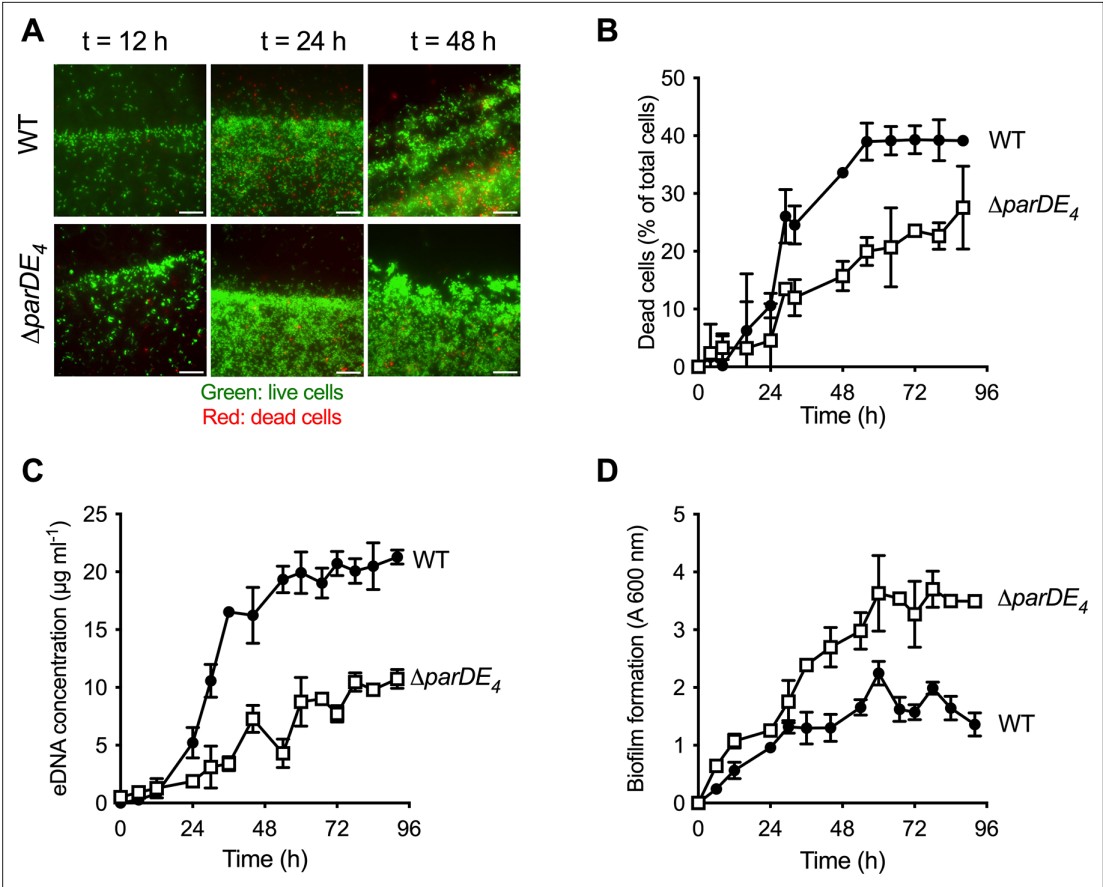

**Figure 2.** Involvement of the ParDE$_4$ TAS in cell death, extracellular DNA (eDNA) release, and biofilm regulation. (**A**) Biofilm formed on polyvinyl chloride (PVC) strips stained with the BacLight Live/Dead reagent at different incubation times. Images represent overlays of the green (live cells) and red (dead cells) signals collected by epifluorescence microscopy. Scale bars = 10 μm. (**B**) Percentage of dead cells over time in the biofilm, calculated from BacLight Live/Dead stained cells using microscopy images. (**C**) eDNA release in the planktonic phase over time, quantified using PicoGreen staining. (**D**) Biofilm formation over time, quantified by staining the attached biomass with crystal violet. *C. crescentus* WT and ΔparDE$_4$ are represented by solid circles and open squares symbols, respectively. Cultures were grown in M2G medium. The results are given as the average of two independent experiments, each run in triplicate, and the error bars represent the standard error of the mean (SEM).

The online version of this article includes the following figure supplement(s) for figure 2:

**Figure supplement 1.** Involvement of the ParDE$_4$ TAS in cell death, extracellular DNA (eDNA) release, and biofilm regulation.

the amount of biofilm formed in the presence of spent media from cultures of different strains. We showed previously that eDNA present in spent medium from saturated cultures inhibits biofilm formation (***Berne et al., 2010***). Spent media, containing various concentrations of eDNA from cultures of either WT or ΔparDE$_4$ grown to late stationary phase were used in biofilm assays, as done previously (***Berne et al., 2010***). Because ΔparDE$_4$ produces less eDNA than WT (~8 μg ml$^{-1}$ compared to ~12 μg ml$^{-1}$ of eDNA present in the spent medium of a saturated culture for ΔparDE$_4$ and WT, respectively, see ***Figure 3—figure supplement 1***), we first determined the amount of eDNA present and added an appropriate amount of spend medium to obtain the same final concentration of eDNA. The amount of biofilm formed by WT and ΔparDE$_4$ was similar for the same total amount of eDNA (***Figure 3—figure supplement 1***), showing that, when exposed to the same amount and source of eDNA, both WT and ΔparDE$_4$ form similar amounts of biofilm and that eDNA from WT or ΔparDE$_4$ have the same biofilm inhibitory activity. Furthermore, the inhibition response is positively correlated with the amount of eDNA present in the spent medium in a similar manner for both strains (***Figure 3—figure supplement 1***), in agreement with our previous results (***Berne et al., 2010***). Therefore, the increase in biofilm formation by the ΔparDE$_4$ mutant is not due to an impaired response to eDNA but is likely due to less cell death causing a more rapid mass increase and/or to less cell dispersal stimulation by eDNA.

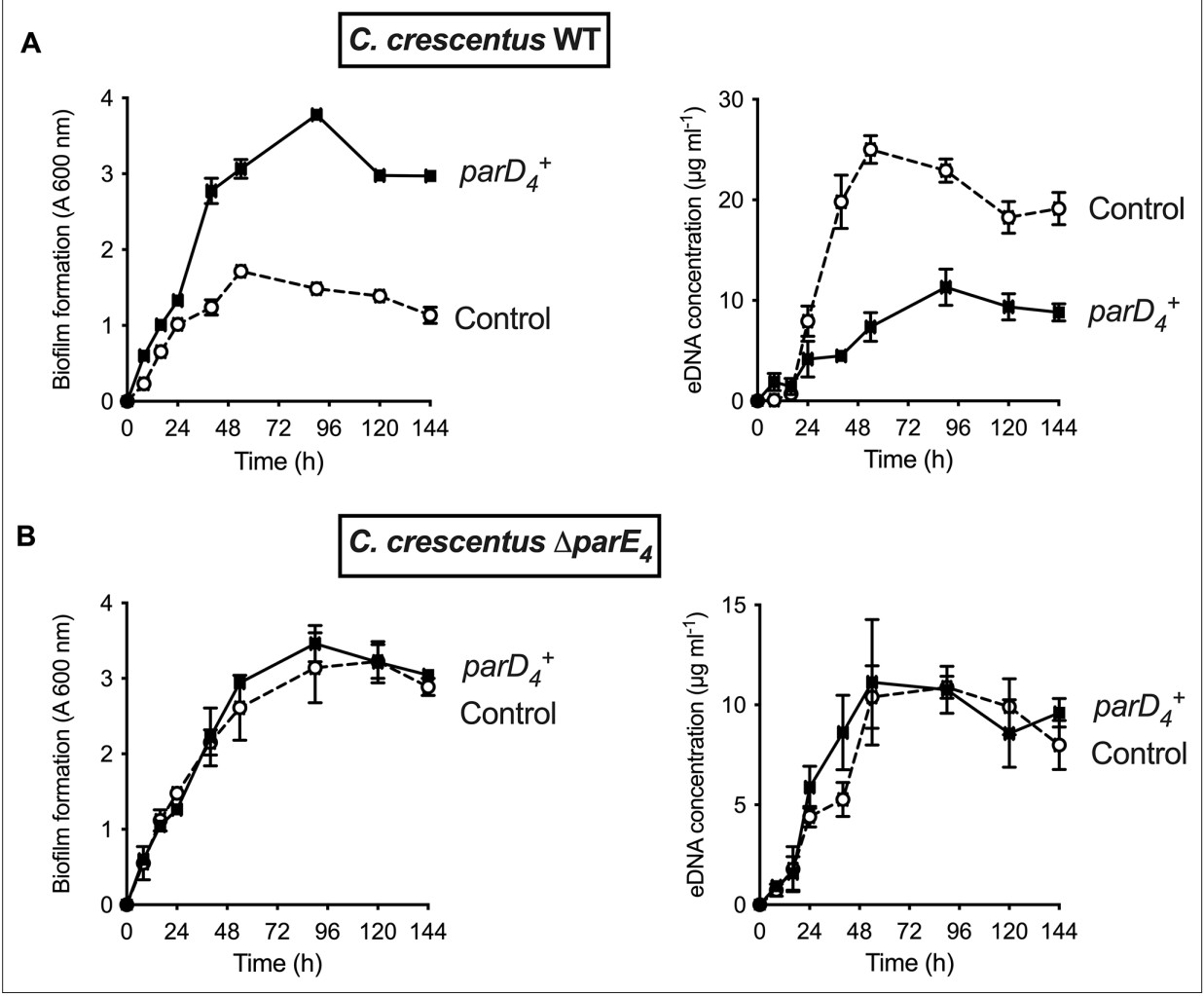

**Figure 3.** Effect of $parD_4$ antitoxin gene-induced constitutive expression on biofilm formation and extracellular DNA (eDNA) release. The $parD_4$ was cloned into the low copy replicating pMT686 plasmid and expressed using the P$xyl$ promoter. Biofilm formation (left panels) and eDNA release (right panels) for strains expressing the $parD_4$ antitoxin gene (solid square symbols) or bearing the empty pMT686 plasmid (open circle symbols) in WT (**A**) and $\Delta parE_4$ (**B**). Cultures were grown in M2G medium + Cm 1 µg ml$^{-1}$. The results are given as the average of three independent experiments and the error bars represent the standard error of the mean (SEM).

The online version of this article includes the following figure supplement(s) for figure 3:

**Figure supplement 1.** Biofilm inhibition in WT and $\Delta parDE_4$ by extracellular DNA (eDNA) from different spent media.

## ParDE$_4$ promotes population dispersal in the biofilm

To understand the dynamics of biofilm formation in the WT and $\Delta parDE_4$ strains, we grew a mixed culture of differently fluorescently labeled WT and $\Delta parDE_4$ at a 1:1 ratio in flow cells. The evolution of each population in the biofilm was monitored over time (*Figure 4A*). Patterns in the spatial organization of the biofilm could be observed at early stages of biofilm maturation (*Figure 4A*), with formation of homogenous microcolonies due to clonal growth. This observation is in agreement with previous reports of *C. crescentus* biofilm growth in flow cells (*Entcheva-Dimitrov and Spormann, 2004*; *Rossy et al., 2019*). While in the early stages of biofilm formation the ratio of WT and $\Delta parDE_4$ was maintained, the mutant population rapidly outcompeted the WT at later stages. After 96 hr, around 80% of the attached bacteria were $\Delta parDE_4$ (*Figure 4B*).

To test the dispersal rate of both strains, we measured dispersal from the biofilm by quantifying the number of cells released from the biofilm over time in the flow cells. This was done by collecting flow-through samples downstream of the flow cell and quantifying the number of single cells released from each population (*Figure 4C*). Surprisingly, while there was more biomass of $\Delta parDE_4$ cells than WT in

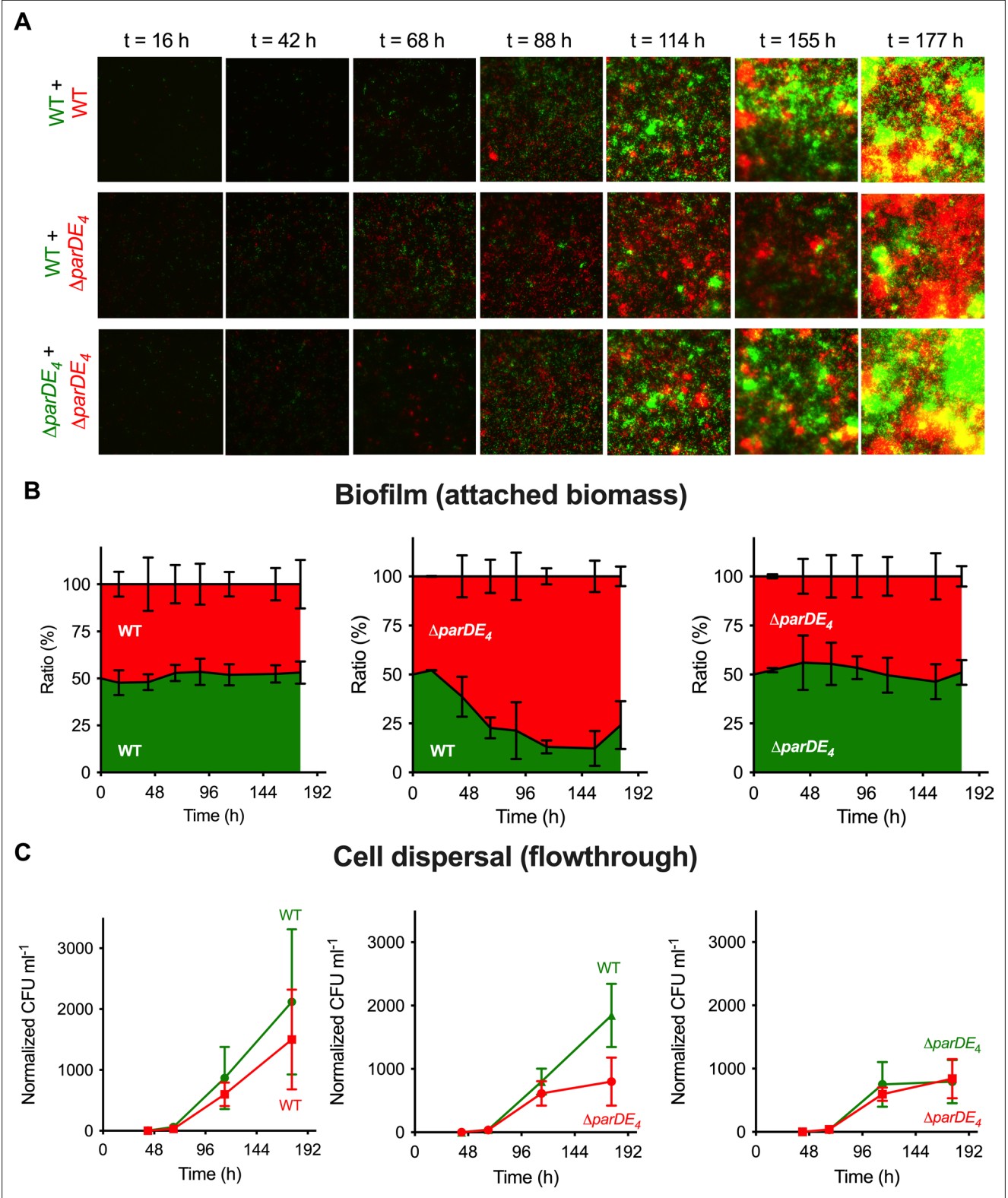

**Figure 4.** Biofilm formation and dispersion in mixed cultures. Differentially fluorescently tagged populations of WT and Δ*parDE₄* were mixed to a 1:1 ratio and grown in flow cells over time. (**A**) Representative fluorescence microscopy images of mixed culture biofilms grown in flow cells. One population is represented in green and the other one in red. (**B**) Ratio of each population over time. Results are given as a percentage of total fluorescent area (green + red) representing both populations, calculated from microscopy images (average of 5 fields of view per time point, during triplicate independent experiments where fluorescent markers were swapped). (**C**) Cell dispersal measured as colony-forming units (CFU ml⁻¹) released in the

*Figure 4 continued on next page*

*Figure 4 continued*

flowthrough downstream of the flow cells. Results are normalized to the number of colonies measured for WT at $t = 42$ hr (beginning of the experiment). Experiments were performed using M2G medium. Results are expressed as an average of 2–4 serial dilution counts of 3 samples per time point. Flow cells were run in triplicate independent experiments where fluorescent markers were swapped. Errors bars represent the standard error of the mean (SEM).

the mixed biofilm experiment, there were more WT cells released over time compared to $\Delta parDE_4$, indicating that the dispersal of WT is more efficient (*Figure 4C*).

In summary, these results combined with those of previous sections suggest that the observed increased biofilm formation in the $\Delta parDE_4$ is due to a combination of increased attached biomass because of reduced cell death and decreased dispersion efficiency.

## The ParDE4 response is correlated with $O_2$ availability

Previous work investigated the regulation of the $parDE_4$ operon as a function of $O_2$ availability on planktonic, exponentially growing cells (*Fiebig et al., 2010*). Transcriptome profiling by microarray performed under hypoxic stress (limiting $O_2$ conditions) showed a twofold decrease in $parD_4$ expression, albeit not statistically significant (*Fiebig et al., 2010*). In our hands, PCD triggered by the $ParDE_4$ TAS is more pronounced when the biofilm reaches maturation (*Figure 2*). A number of environmental changes occur as the biofilm matures, including variation of $O_2$ availability. Since *C. crescentus* is an obligate aerobe, $O_2$ depletion could be a major detrimental factor, triggering cell death and dispersal, as is the case in other species (*McDougald et al., 2011*).

In order to test the effect of $O_2$ on $ParDE_4$-mediated cell death, we grew cells with vigorous shaking ('maximal aeration') as compared to the non-shaking 'moderate aeration' condition (*Figure 5A*). Interestingly, a decrease in $\Delta parDE_4$-mediated eDNA release did not occur in cells grown under vigorous shaking (maximal aeration conditions) compared to the moderate aeration conditions (*Figure 5B* and *Figure 1—figure supplement 1*), suggesting that $ParDE_4$ is not active under those conditions. To determine if $ParDE_4$ expression is regulated by aeration conditions, we monitored its transcription using a *lacZ* fusion under maximal aeration growth compared to growth under moderate aeration conditions. To validate those growth conditions as providing different $O_2$ availability, we used a *lacZ* fusion to the promoter of *ccoN*, encoding the cytochrome $cbb_3$ oxidase subunit I. This gene is highly expressed when *C. crescentus* cells experience limiting $O_2$ levels (*Crosson et al., 2005*) and its expression can be used as a biosensor to monitor $O_2$ availability. P*ccoN* expression was 10–13 times more active under moderate aeration conditions (*Figure 5C*), confirming that $O_2$ availability is limited under those growth conditions. We found that P*parDE_4* transcription was approximately two to three times higher under maximal aeration growth (*Figure 5C*), suggesting that $ParDE_4$ expression is regulated by $O_2$ availability.

Since the above results suggested that $O_2$ availability is important for $ParDE_4$-controlled biofilm regulation, we tested two additional growth conditions to obtain an intermediate level of $O_2$ ('high aeration') and a hypoxic level ('limited aeration') (*Figure 6A*), as assessed by measuring the *ccoN* transcript levels in each culture (*Figure 6B*). We first confirmed that the expression of P*parDE_4* is inversely correlated to the $O_2$ level in the cultures (*Figure 6B*). We then compared the ratio of eDNA released in $\Delta parDE_4$ relative to WT in the different aeration conditions. We found that there was twice as much eDNA released by WT compared to $\Delta parDE_4$ under conditions where $O_2$ levels are the most reduced whereas the levels were similar under maximal aeration (*Figure 6C*). Concomitant with the decrease in eDNA release, we saw that biofilm formation by the $\Delta parDE_4$ increased relative to WT as $O_2$ availability decreased (*Figure 6D*). In summary, lower $O_2$ availability correlated with less eDNA release and more biofilm formation by the $\Delta parDE_4$ strain compared to WT. We therefore conclude that $ParDE_4$ triggers cell death when $O_2$ is limited, which in turn releases eDNA. As shown in the previous sections and in our previous work (*Berne et al., 2010*), eDNA inhibits biofilm formation.

Next, we wanted to determine if $O_2$ limitation and $parDE_4$ expression are correlated at the single cell level within a population. For that purpose, we created a fluorescent reporter fusion for the $parDE_4$ operon by fusing its promoter region to a promoterless GFP construct to monitor $parDE_4$ expression via GFP fluorescence signal. We also constructed a mCherry fusion to the promoter region of *ccoN* to monitor $O_2$ availability for each cell using mCherry fluorescence as a proxy. We grew cells carrying both reporters under our four growth conditions with different $O_2$ availability (*Figure 6A*). We

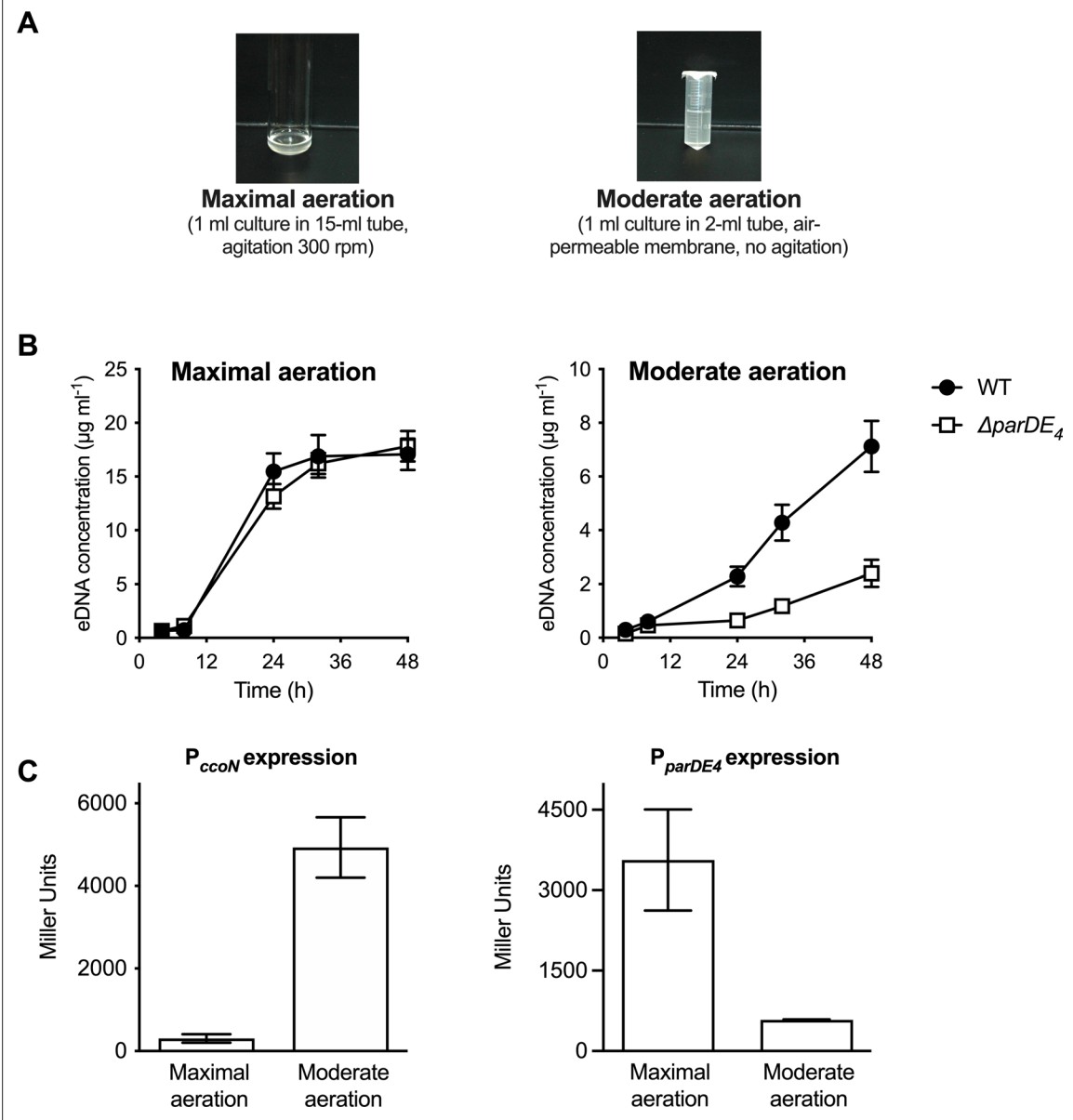

**Figure 5.** P*parDE₄* expression is induced under maximal aeration growth conditions. (**A**) Images of cultures grown in M2G under conditions providing different amounts of $O_2$, termed maximal and moderate aeration, respectively. (**B**) Quantification of extracellular DNA (eDNA) released in the planktonic phase of WT (solid circles) and Δ*parDE₄* (open squares), quantified using PicoGreen staining. Results are given as the average of five independent experiments and the error bars represent the standard error of the mean (SEM). (**C**) β-Galactosidase activity of P*parDE4*-*lacZ* (right) and P*ccoN*-*lacZ* transcriptional (left) fusions in WT grown under maximal and moderate aeration conditions (as illustrated in panel A). The results represent the average of six independent cultures (assayed on three different days) and the error bars represent the SEM.

first quantified the number of GFP and mCherry expressing cells under each condition (***Figure 7A***). As expected from the activity of the *ccoN* promoter, the number of mCherry expressing cells decreased as growth conditions became less anoxic: while more than 60% of cells expressed mCherry under limited aeration, only 6% did so under maximal aeration (***Figure 7A***). In contrast, we observed an opposite trend for GFP expression driven by the *parDE₄* promoter; it increased with increasing $O_2$ availability in the culture, with more than 40% of cells expressing GFP under maximal and high aeration conditions and around 10% under moderate and limited aeration (***Figure 7A***). We also monitored the red and green fluorescence intensity per cell in the four different growth conditions (***Figure 7B*** and ***Figure 7—figure supplement 1***). There was a strong anticorrelation between the expression of *ccoN* and *parDE₄* in single cells. In other words, cells which experienced hypoxia rarely expressed

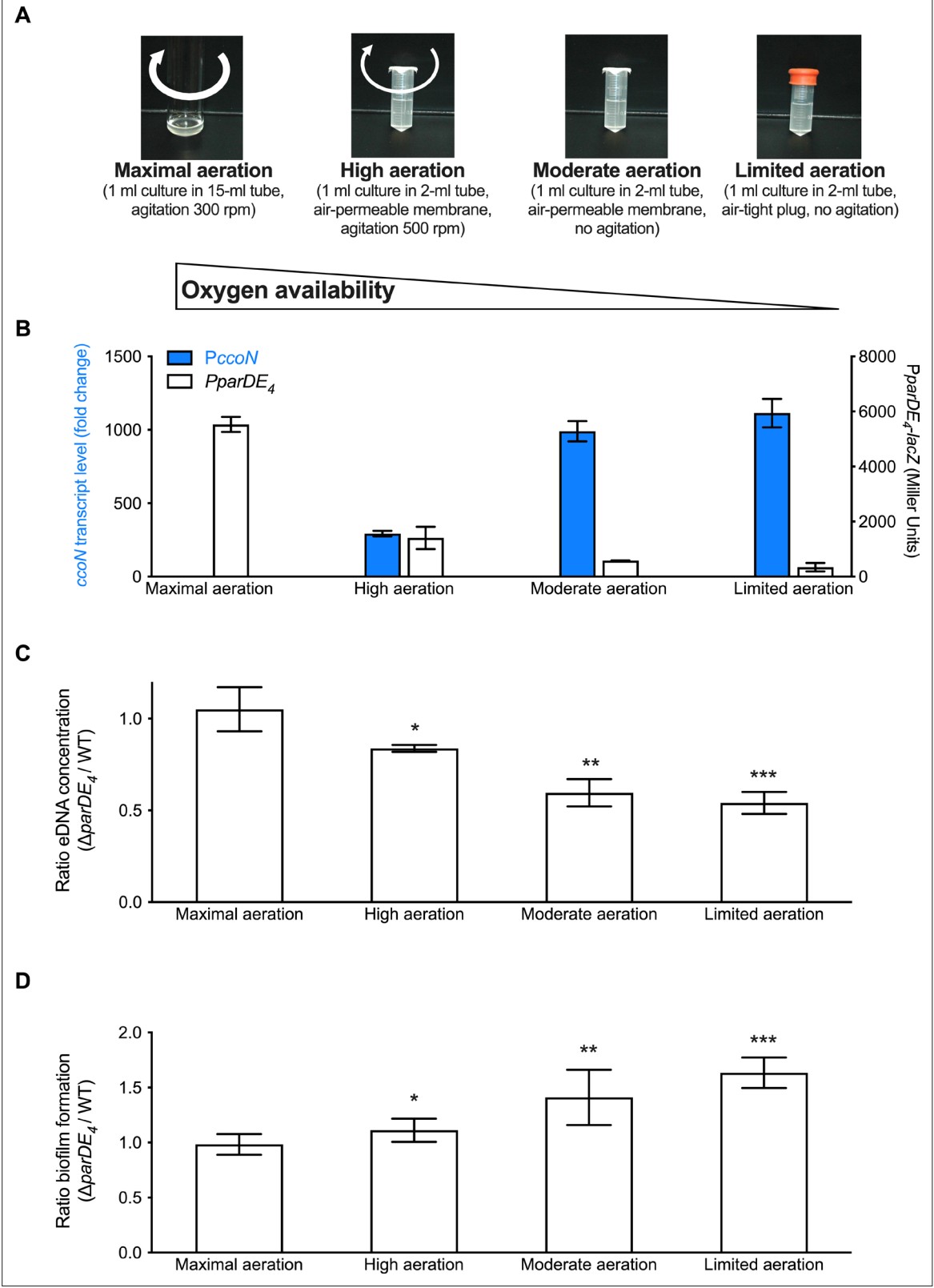

**Figure 6.** Extracellular DNA (eDNA) release and biofilm formation under variable $O_2$ availability. (**A**) Images of M2G-grown cultures providing different amounts of $O_2$, termed maximal, high, moderate, and limited aeration, respectively. (**B**) Assessment of P*parDE4* and P*ccoN* expression (white and blue bars, respectively) by measuring β-galactosidase activity of the P*parDE4-lacZ* transcriptional fusions in WT, and transcription of *ccoN* relative to *rpoD* as a function of various aeration conditions by RT-qPCR in the same cultures. qPCR results (calculated as described in the Material and method section)

*Figure 6 continued on next page*

*Figure 6 continued*

were normalized to the results in 'Maximal aeration' conditions set to 1. (**C**) Ratio of the eDNA release concentration measured in ΔparDE4 planktonic phase over WT, in cultures grown under the different aeration conditions. Results are given as the calculated ratio of the eDNA concentration measured in six independent replicates, each run in duplicate, and error bars represent the standard error of the mean (SEM). Statistical comparisons to 'maximal aeration' conditions are calculated using Student's unpaired t-tests; *p < 0.5, **p < 0.05, ***p < 0.005. (**D**) Biofilm formation after 24 hr when cells are grown under the different aeration conditions. Results are given as the calculated ratio of the amount of biofilm formed in three to five independent replicates, each run in triplicate, and error bars represent the SEM. Statistical comparisons to 'maximal aeration' conditions are calculated using Student's unpaired t-tests; *p < 0.05, **p < 0.01, ***p < 0.005.

*parDE4*, while cells sensing higher levels of $O_2$ levels did. These results confirm what we previously observed at the population level, that is that *parDE4* expression is correlated with $O_2$ availability.

## ParDE4 is required for increased cell death in biofilm areas with lower $O_2$ availability

To assess how local amounts of $O_2$ might influence ParDE4 controlled cell death within an established biofilm, we next assessed the pattern of cell death and *parDE4* expression in different areas of the biofilm. We placed a PVC strip inside Aeraseal covered microtubes. We monitored three distinct areas: (1) the air–liquid interface area at the meniscus (highest air exchange); (2) the middle of the strip (moderate air exchange); and (3) the bottom of the strip (limited air exchange) (*Figure 8*). In each area, we determined the *parDE4* or *ccoN* promoter activity using a P*parDE4*-*lacZ* or P*ccoN*-*lacZ* transcriptional fusion, respectively, and fluorescein Di-β-D-Galactopyranoside (FDG), a fluorogenic substrate whose fluorescence is turned on upon cleavage by β-galactosidase (*Rotman et al., 1963*). When we monitored *parDE4* activity, we could measure a decrease in fluorescence for cells attached at the bottom of the biofilm compared to the meniscus or the middle of the strip, indicating that the P*parDE4*-*lacZ* reporter is less active in areas with lower $O_2$ availability (*Figure 8A*, left panels). P*ccoN*-*lacZ* activity showed that cells at the bottom of the coverslips

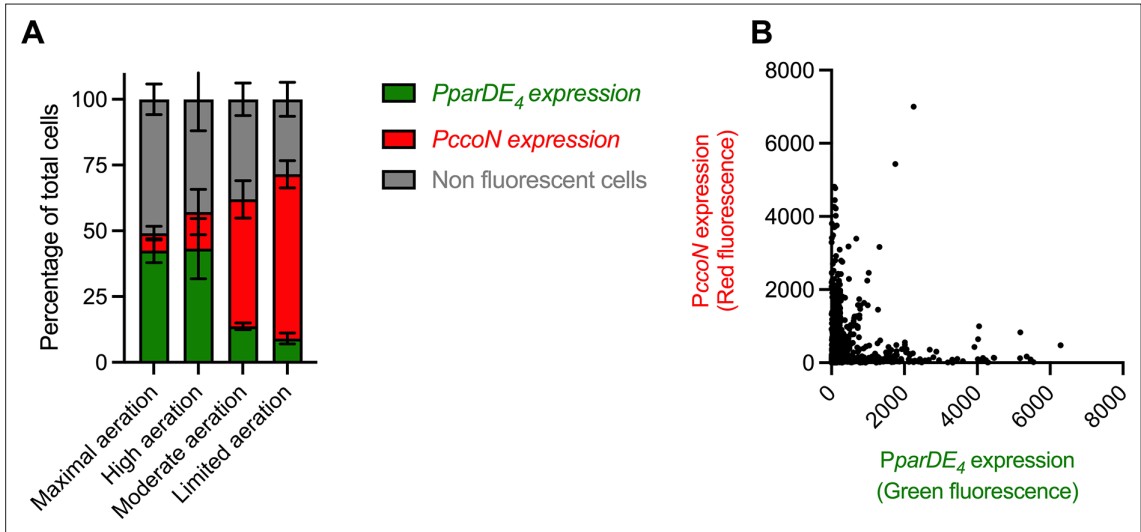

**Figure 7.** Expression of the *parDE4* operon is anticorrelated with a hypoxia reporter. (**A**) Number of cells expressing GFP (P*parDE4* expression) and mCherry (P*ccoN* expression) in the whole population when cells were grown in M2G under maximal, high, moderate, and limited aeration, providing different amounts of $O_2$. A cell was considered expressing GFP and/or mCherry if the fluorescence signal was at least 1.2 times the background. The error bars represent the standard error of the mean (SEM). (**B**) Red and green fluorescence intensity of single cells grown under maximal, high, moderate, and limited aeration (all conditions combined). WT cells carrying both pMR20-P*parDE4*-*gfp* and pMR10-P*ccoN*-*mcherry* plasmids were grown to $OD_{600}$ = 0.4–0.6 under maximal, high, moderate, and limited aeration (*Figure 6A*) and imaged by epifluorescence microscopy. More than 3000 cells from at least three independent replicates were quantified for the number of cells with a green or red fluorescent signal (**A**) and the intensity of these fluorescent signals (**B**).

The online version of this article includes the following figure supplement(s) for figure 7:

**Figure supplement 1.** Expressions of the *parDE4* operon is reduced when a hypoxia reporter expression is increased.

**Figure supplement 2.** Cell morphology is not impacted by hypoxia.

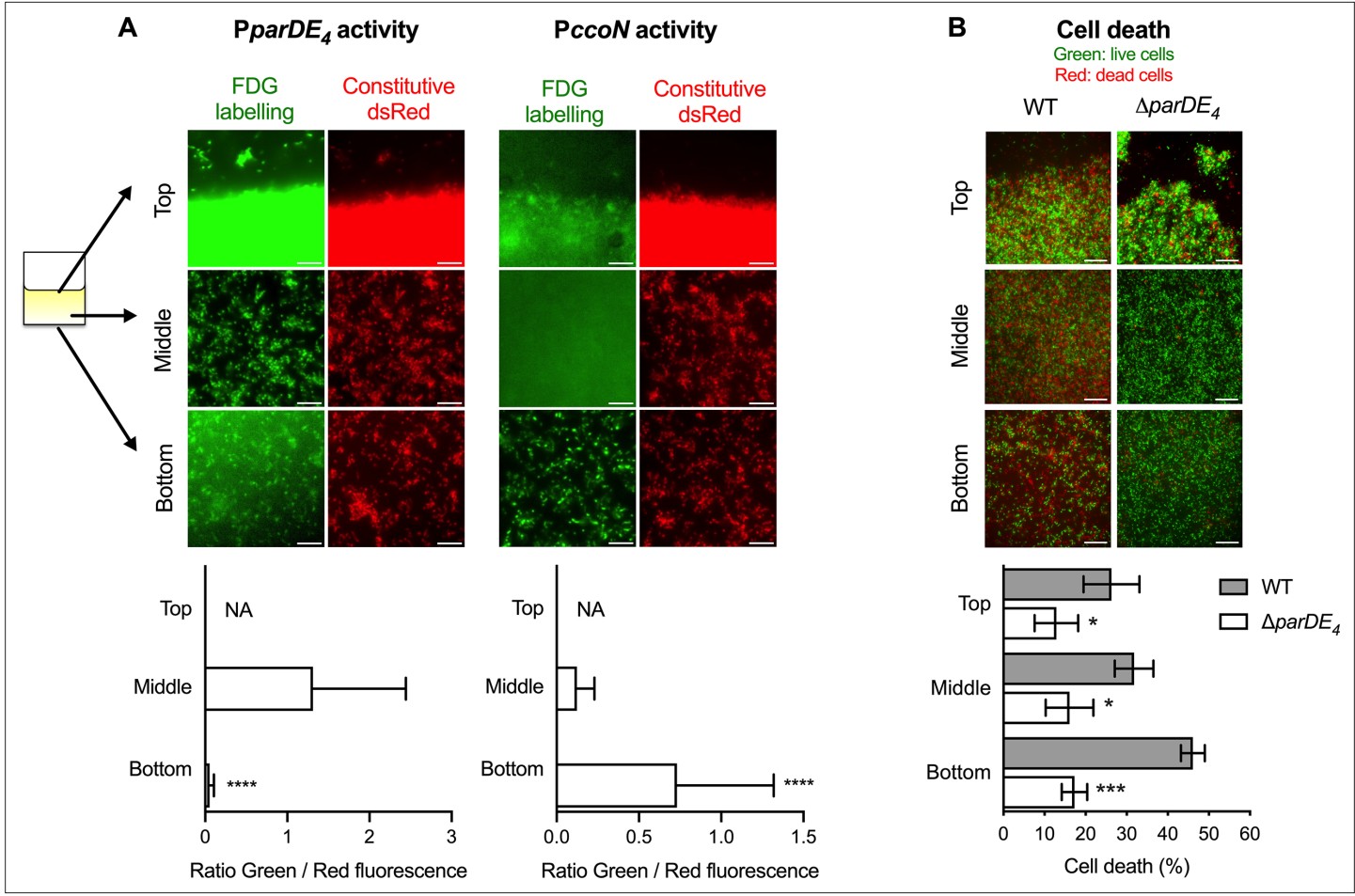

**Figure 8.** Spatial regulation of *parDE₄* expression and ParDE₄-mediated cell death in the biofilm. Biofilms were grown on polyvinyl chloride (PVC) strips in M2G medium. Three locations were monitored: the air–liquid interface (Top), the middle of the strip (Middle), and the bottom of the strip (Bottom). (**A**) P*parDE₄* and P*ccoN* activities in cells attached at the three different locations. Red fluorescent WT cells (WT::miniTn7-*dsred*) carrying either the P*parDE₄-lacZ* or the P*ccoN-lacZ* plasmids were grown for 36 hr, and promoter activities were measured by the amount of fluorescein cleaved by the β-galactosidase from the fluorogenic FDG substrate. While representative fluorescence images are shown using the same set brightness/contrast for the red channel, brightness/contrast are set automatically in ImageJ for each green fluorescence image. Scale bars = 10 µm. Green (FDG labeling, promoter activity) and red (Constitutive Dsred signal, cell density) fluorescence intensities were quantified using microscopy images of 10 fields of views of three independent replicates (more than 10,000 events). Results are expressed as a ratio of green/red fluorescence signals and are shown as averages with standard deviation (SD). Statistical comparisons of the ratio calculated at the bottom of the coverslips to the middle are calculated using paired *t*-tests, ***p < 0.0001. We could not determine (not available, NA) the amount of FDG hydrolyzation at the air–liquid interface, as the biofilm was too dense to accurately quantify fluorescence signals due to saturation. (**B**) Cell death quantification of biofilms of WT and Δ*parDE₄* after 36 hr. Percentage of dead cells over time in the biofilm, calculated from BacLight Live/Dead stained cells using microscopy images. Results are given as the average of percentage of dead cells (red) in 10 fields of views of three independent replicates and are shown in gray and white bars for WT and Δ*parDE₄*, respectively. Error bars represent standard error of the mean (SEM). Statistical comparisons to WT in the same condition are calculated using Student's unpaired *t*-tests; *p < 0.05, ***p < 0.005, ****p < 0.0005.

experience hypoxia, as more fluorescence could be detected, indicating that *ccoN* expression is induced in this environment compared to the middle of the coverslip or the meniscus (**Figure 8A**, right panels).

When we monitored cell death in WT and Δ*parDE₄* biofilms grown in the same manner, we found that WT displayed more cell death away from the meniscus, showing that cell death preferentially occurs where $O_2$ is limited (**Figure 8B**). Indeed, twice as many cells were dead at the bottom of the strip compared to the top (**Figure 8B**). However, in the Δ*parDE₄* mutant, the amount of cell death was not significantly different between the distinct locations within the biofilm, indicating that the ParDE₄ TAS regulates PCD induced by $O_2$ limitation.

## Discussion

While eDNA has been shown in the past to be a major component of the biofilm matrix that plays a role in overall biofilm architecture for different species (*Okshevsky and Meyer, 2015*; *Campoccia et al., 2021*), we previously reported a novel role for this molecule, as an environmental cue to sense deleterious environments in a biofilm and promote cell dispersion in *C. crescentus* (*Berne et al., 2010*). We showed that cell death and eDNA release increase during biofilm formation and that eDNA prevents newborn swarmer cells from attaching to surfaces and settling into a biofilm by binding specifically to and inhibiting the adhesiveness of the holdfast. Because stalked cells attached by their holdfast cannot be dislodged from a surface by eDNA, this mechanism promotes swarmer cell dispersal without causing a potentially undesirable dissolution of the existing biofilm. While our previous results support the role of eDNA as a cue that prevents settling of swarmer cells (*Berne et al., 2010*; *Kirkpatrick and Viollier, 2010*), it was not known if this mechanism was simply a consequence of random cell death or the result of an active mechanism.

In this work, we show that *C. crescentus* biofilms experience increasing cell death in regions of a mature biofilm where $O_2$ becomes limiting. We show that this increased cell death in biofilm regions of $O_2$ limitation depends on the *parDE4* TAS. We also show that *parDE4*-dependent cell death is highest in conditions where $O_2$ is limited. Under conditions of maximal aeration, cell death and eDNA release are equivalent in WT and in the *parDE4* mutant, whereas they are higher in WT under conditions of limited aeration.

Interestingly, even if cells of the *parDE4* mutant increase their biofilm biomass more rapidly than WT, they have a reduced dispersal from the biofilm. This result is consistent with our previous findings that eDNA release increases as biofilms mature and that eDNA inhibits swarmer cell adhesion by binding to their holdfast, but does not dislodge previously bound cells (*Berne et al., 2010*). We

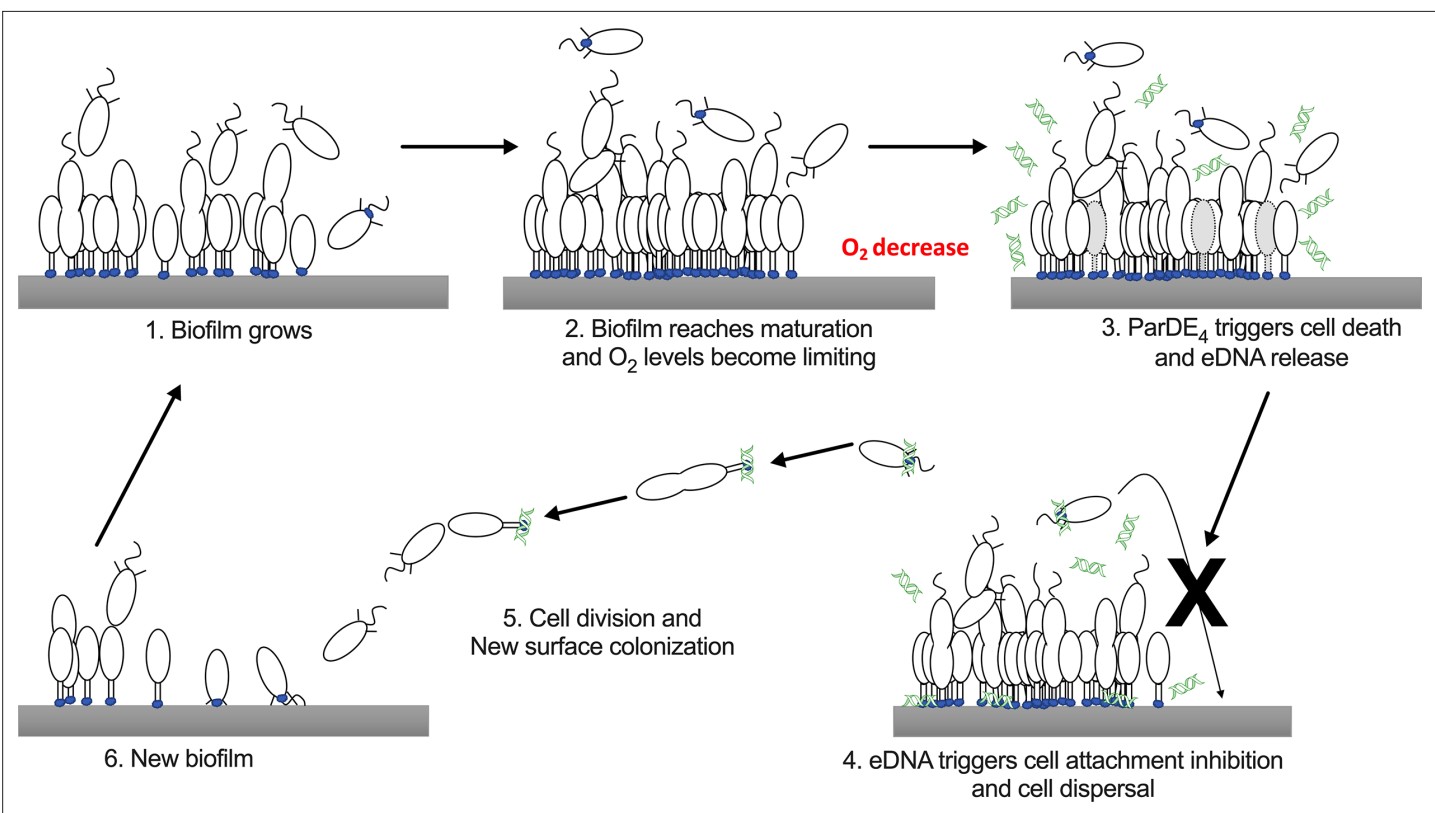

**Figure 9.** Schematic representation of the hypothetical mechanism of ParDE4-dependent regulation of cell death and dispersal upon $O_2$ limitation. Once the biofilm reaches maturation, $O_2$ availability becomes limiting. ParDE4-mediated programmed cell death (PCD) is initiated upon $O_2$ deprivation and targeted cells release extracellular DNA (eDNA) via cell lysis. eDNA specifically binds to holdfasts, preventing new cells from attaching, but does not influence already attached cells. Unable to join the biofilm, swarmer cells disperse, divide, and their offspring eventually find a new surface to colonize.

suggest that, as the *C. crescentus* biofilm increases in biomass, $O_2$ limitation activates the $ParDE_4$ TAS, resulting in cell death and concomitant eDNA release, which promotes dispersal of swarmer cells by binding to their holdfast and inhibiting their adhesion. This process enables the colonization of new environments while preserving the survival of the initial biofilm (*Figure 9*).

We showed that both WT and Δ*parDE₄* strains react in a similar manner to the presence of excess eDNA (*Figure 3—figure supplement 1*), so how can we reconcile this with our observation that WT cells disperse more efficiently than Δ*parDE₄* cells (*Figure 4C*)? *C. crescentus* biofilms are composed of discrete, clonal microcolonies (*Figure 4A* and *Entcheva-Dimitrov and Spormann, 2004*; *Rossy et al., 2019*). These microcolonies provide a microenvironment where conditions can differ from other microcolonies in the same biofilm. As cells die in biofilm microcolonies, the released eDNA will be at maximum concentration in the microcolony where it was released and will interact with nearby cells before diffusing to adjacent microcolonies where its concentration will be lower. Since eDNA inhibition of swarmer cell adhesion is proportional to its concentration, microcolonies with increased death rate will contain more eDNA and have a higher swarmer cell dispersal rate. Cells from WT microcolonies, which have a higher death rate in mature biofilms, should therefore have a higher rate of dispersal than cells from Δ*parDE₄* microcolonies.

In addition to promoting dispersal, the PCD mechanism we describe here may also help preserve the initial biofilm. Indeed, once irreversibly attached via their holdfasts, stalked cells cannot disperse (*Berne et al., 2010*). In addition to stimulating dispersal of swarmer cells, PCD of some of the stalked cells in the biofilm may help reduce overcrowding, $O_2$ starvation, and the death of the entire colony. Moreover, $O_2$ limitation can result not only from overcrowding but also from environmental changes. Since *C. crescentus* is an obligate aerobe, this PCD mechanism can also act as a safety trigger that promotes the propagation of daughter cells to more hospitable environments when the colony starts to experience suboptimal $O_2$ levels resulting from an external modification of the environment. Since the magnitude of inhibition of swarmer cell adhesion is proportional to eDNA concentration, this PCD mechanism may act as a rheostat tying cell dispersal to the availability of $O_2$. Other PCD mechanisms in *C. crescentus* could similarly tie cell dispersal from the biofilm to other environmental conditions.

The involvement of PCD in biofilm regulation is well documented in different bacterial species and different TAS have been reported to play a role in biofilm regulation and cell dispersal (*Wang and Wood, 2011*; *Wen et al., 2014*; *Kamruzzaman et al., 2021*; *Singh et al., 2021*). In *Escherichia coli*, several TAS are involved in regulation of biofilm formation at different stages (*Kim et al., 2009*; *Kolodkin-Gal et al., 2009*; *Yamaguchi and Inouye, 2011*; *Soo and Wood, 2013*; *Zhao et al., 2013*). Other TAS have been reported to regulate biofilms in different species, such as in *Vibrio cholerae* (*Wang et al., 2015*), *Staphylococcus aureus* (*Kato et al., 2017*), or *Pseudomonas aeruginosa* (*Wood and Wood, 2016*; *Shmidov et al., 2022*; *Song et al., 2022*). $O_2$ limitation is one of the major environmental stresses experienced by bacteria in a mature biofilm (*Stewart and Franklin, 2008*; *Kostakioti et al., 2013*) and some species, such as *Shewanella oneidensis* or *P. aeruginosa*, respond to $O_2$ depletion by triggering single cell dispersion (*Thormann et al., 2005*; *Barraud et al., 2009*; *Rumbaugh and Sauer, 2020*). PCD is involved in biofilm regulation upon $O_2$ limitation in *S. aureus*, via the coordinated increased expression of the AltA murein hydrolase and decreased expression of the cell wall-teichoic acids (*Mashruwala et al., 2017*). In addition, the Hha/TomB TAS regulates cell death in *E. coli* biofilms in response to $O_2$ availability (*Marimon et al., 2016*): under high $O_2$ conditions, the toxin is inactivated by spontaneous oxidation enhanced by transient interactions with the antitoxin under those conditions. In $O_2$-limited regions of the biofilm, the TomB anitoxin fails to efficiently oxidaze the Hha toxin stimulates cell death and enables the formation of gaps in the biofilm matrix, thereby facilitating biofilm dispersal through those gaps (*García-Contreras et al., 2008*; *Marimon et al., 2016*). However, to our knowledge, none of the previously reported systems function as described in this study.

We showed that the $O_2$-limited conditions where *parDE₄*-dependent cell death occurs correspond to the lowest level of *parDE₄* transcription. This result suggests the possibility that lowering the expression of the *parDE₄* operon triggers cell death. During surface adhesion and in early stages of biofilm development, $O_2$ availability is good relative to later stages. When biomass increases, $O_2$ becomes limiting, and cell death is increased. It is interesting to note that the 'neutralized' steady state of the $ParDE_4$ TAS, when the toxin is inactivated, seems to be when $O_2$ is abundant, that is, when *parDE₄* transcription is at its highest. In most studied TAS, stresses have been shown to induce transcription of TAS (*LeRoux et al., 2020*; *Jurėnas et al., 2022*), but here, the stress inflicted on the cells by $O_2$

limitation is accompanied by a lower expression of *parDE₄*. We are unaware of cases where reduced TAS expression is correlated with the condition that activates the PCD in biofilm regulation. This suggests a novel regulatory mechanism of PCD, in the context of biofilms, that remains to be understood. The system described in this study is conceptually similar to TAS involved in plasmid maintenance/postsegregational killing or phage defence mechanisms. Indeed, TAS encoded on a plasmid, such as the CcdAB (*Ogura and Hiraga, 1983*) or Hok/Sok (*Gerdes et al., 1985*) pairs in *E. coli*, ensure plasmid maintenance over generations: if a daughter cell fails to inherit the plasmid bearing the TAS upon division, the amount of labile antitoxins in this cell quickly drops, allowing free toxins to kill. A similar strategy is seen in instances of phage neutralization upon infection (*LeRoux and Laub, 2022*) by TAS such as ToxIN (*Fineran et al., 2009*) and DarTG (*LeRoux et al., 2022*) for example. Phage infection usually provokes inhibition of the host cell transcription, followed by degradation of the antitoxins and liberation of the toxins. Infected cells are rapidly killed, preventing massive phage replication. In this example, killing a few cells via PCD neutralizes phage infection and rescues the rest of the colony. We propose that the ParDE₄-mediated PCD we describe in this study might play a similar role: selectively killing a few cells in a crowded biofilm where conditions are detrimental to promote the dispersal of newborn cells from the deleterious conditions.

TAS have often been reported to play a direct and crucial role in adaptation to different stresses. However, the physiological role of TAS-regulated cell death has been challenged in recent years (*Harms et al., 2017*; *Goormaghtigh et al., 2018*; *Wade and Laub, 2019*; *Fraikin et al., 2020*; *Rosendahl et al., 2020*). Even if a TAS operon is transcribed under stress conditions, it does not necessarily mean that the toxin is liberated, and transcription might not always reflect TAS activity (*LeRoux et al., 2020*). Therefore, whether the decrease in *parDE₄* transcription that correlates with increased cell death is mechanistically related to TA activation is still not known.

Our future work will focus on identifying the pathway that controls the activation of the ParDE₄-mediated PCD. Based on sequence similarity, ParDE₄ belongs to the type II TAS family (*Fiebig et al., 2010*). In *C. crescentus*, the ParDE₄ homolog ParDE₁ is thought to have a bacteriostatic effect, inhibiting cell division and causing cell filamentation by an unknown mechanism (*Fiebig et al., 2010*). Under conditions where ParDE₄-correlated cell death is observed (O₂-limited conditions), cell morphology does not change significantly (*Figure 8* and *Figure 7—figure supplement 2*). In *E. coli*, the plasmid borne ParDE TAS inhibits DNA replication by targeting DNA gyrase, leading to cell death (*Jiang et al., 2002*). However, the *E. coli* ParE₂ toxin, whose structure is close to that of the predicted *C. crescentus* ParE₁ (*Dalton and Crosson, 2010*) does not bind to DNA gyrase and the molecular target of this toxin is still unknown (*Sterckx et al., 2016*). These results suggest that the ParDE family is not homogenous in terms of targets, mechanisms, and physiological outcomes. So far, we have failed to express the *parE₄* gene both in *E. coli* and *C. crescentus*, suggesting that its product is highly toxic. Our future efforts will aim at engineering an attenuated version of *parE₄*. Such a tool will help to identify its target(s) and elucidate the pathway that leads to cell killing by this TAS.

Does the PCD-stimulated cell dispersal mechanism we have described constitute cooperation and even altruism? Altruism mechanisms in biofilms have been discussed previously (*Penesyan et al., 2021*; *Sadiq et al., 2021*). Altruism involves a trade-off between growth rate and growth yield in the biofilm to preserve resources at the cost of individual growth rate for the fitness of the entire population (*Kreft, 2004*), or the production of public goods that are used by the whole attached colony (*Nadell et al., 2009*). In addition, to be beneficial for the overall community while being costly for the individual, Hamilton's rule states that altruism must promote only the success of the surviving kin (*Hamilton, 1970*; *Ramisetty and Sudhakari, 2020*). We showed previously that eDNA stimulation of dispersal through holdfast inhibition is highly specific to *Caulobacter* species: biofilms of other tested species were not impaired by addition of eDNA, including *Caulobacter* DNA, and *Caulobacter* adhesion is only inhibited by eDNA from *Caulobacter* (*Berne et al., 2010*). Our current results show that deletion of ParDE₄ decreases dispersal from the mature biofilm. Intuitively, stimulating dispersal away from a biofilm where environmental quality is declining to another, potentially better environment would appear to be advantageous for the dispersing kin in terms of fitness, but this has not been directly demonstrated in our case. Reducing the competing population in a mature biofilm with declining O₂ availability could also provide an advantage to the surviving attached cells, but this is difficult to determine experimentally. Thus, the PCD-mediated mechanism described in this study

might confer a fitness advantage to the overall *Caulobacter* population under the appropriate environmental conditions.

## Materials and methods

### Bacterial strains and growth conditions

All bacterial strains used in this study are listed in *Supplementary file 1*. *Escherichia coli* strains used for cloning experiments were grown in LB medium at 37°C. When necessary, antibiotics were added at the following final concentrations: chloramphenicol (Cm) 50 µg ml⁻¹ (for pMT686 constructs), and tetracycline (Tet) 15 µg ml⁻¹ (for pRKlac290 constructs). *C. crescentus* strains were grown at 30°C using M2 minimal medium complemented with 0.2% glucose (M2G) (*Johnson and Ely, 1977*) in liquid, and using Peptone Yeast Extract (PYE) (*Poindexter, 1964*) + 15 g l⁻¹ bactoagar (Difco) plates. All experiments were performed using M2G, while *C. crescentus* were grown in PYE for strain construction. *C. crescentus* cultures were grown under four different aeration conditions: (1) maximal aeration by growing 1 ml of culture in a 15-ml glass culture tube tilted and shaken at 300 rpm; (2) high aeration with 1 ml of culture in a 2-ml microtube sealed with a 1.5 × 1.5 cm piece of sterile breathable sealing film (AeraSeal, Excel Scientific), to allow air exchange, in a shaking ThermoMixer (Eppendorf), at 500 rpm; (3) moderate aeration with 1 ml of a culture in a 2-ml microtube sealed with a 1.5 × 1.5 cm piece of sterile breathable sealing film (AeraSeal, Excel Scientific), to allow air exchange, in a non-shaking ThermoMixer (Eppendorf); and (4) limited aeration with 1 ml of a culture in a 2-ml microtube tightly sealed with an air tight rubber plug to prevent air exchange with the environment, in a non-shaking ThermoMixer (Eppendorf). When appropriate, Cm and Tet were added to 1 and 2 µg ml⁻¹ when using plasmids pMT686 and pRKlac290, respectively. To grow the pMR10-P*ccoN-mcherry* pMR20-P*parDE₄-gfp* harboring strain, both Kan and Tet were added to 0.5 µg ml⁻¹.

*C. crescentus* Δ*parDE₄* strains harboring the stable miniTn7*gfp* or miniTn7*dsred* (CB15 Δ*parDE₄*::miniTn7*gfp* (YB5253) and CB15 Δ*parDE₄*::miniTn7*dsred* (YB5254)) were constructed by φCr30-mediated transduction (*West et al., 2002*) from AS110 and AS109 (*Entcheva-Dimitrov and Spormann, 2004*), respectively, into *C. crescentus* CB15 Δ*parDE₄* (FC915).

### Plasmid construction and cloning procedures

All plasmids were cloned using standard molecular biology techniques. PCR were performed using *C. crescentus* CB15 WT (YB135) gDNA as the template. Sequences of the primers used are described in *Supplementary file 2*.

To construct a *parD₄*-inducible expression plasmid, the entire *parD₄* gene was cloned in frame into the low copy xylose-inducible replicating plasmid pMT686 (*Thanbichler et al., 2007*), to give plasmid pMT686-*parD₄* (*Fiebig et al., 2010*). To construct a transcriptional fusion plasmid for the *parDE₄* promoter, 567 bp upstream of the *parD₄* start codon were cloned into pRKlac290 (*Gober and Shapiro, 1992*) upstream and in the frame of *lacZ*, to give plasmid pRKlac290-P*parDE₄*.

### Live/Dead quantification in planktonic cultures

Cell death in planktonic cultures was detected using the BacLight Bacterial Viability kit (L7007, Invitrogen) and quantified by fluorimetry. A ratio of 1:1 Live/Dead stain mixture from the kit was diluted to 1/1000 in sterile dH₂O. One hundred µl of diluted Live/Dead stain were added to 100 µl of the liquid culture to be assayed. The resulting 200 µl were pipetted to a black 96-well plate with a clear flat bottom (Corning) prior to measurements using a Synergy HT microplate reader. Standard samples were added to each plate of samples, to generate a calibration curve that allows experimental samples quantification. Mid-log phase *C. crescentus* cells grown in M2G were diluted to an OD₆₀₀ of 0.1. The diluted culture was divided into two samples: one sample was mixed with an equal volume of M2 medium (with no carbon source) (*Johnson and Ely, 1977*), to become the live cell standard sample; and one sample was mixed with an equal volume of 70% isopropyl alcohol, centrifuged twice and resuspended in M2, to become the dead cell standard sample. Different ratios of live and dead cell samples were mixed to provide five standards (ratio live:dead of 100:0, 75:25, 50:50, 25:75, 0:100). Absorbance at 600 nm (A₆₀₀), green and red fluorescence signals (using 485/528 and 485/630 nm Em/Ex filters, respectively) were quantified for each well. Data were analyzed by calculating the ratio of fluorescence emission at 528 nm (green signal)/fluorescence emission at 630 nm (red signal). The

values calculated for the standard samples were plotted versus the percentage of live cells in those standards, and a linear regression from the standard curve was used to determine the percentages of live and dead cells in each sample.

### eDNA quantification

eDNA was quantified using the Quant-iT PicoGreen reagent (Invitrogen), as described previously (*Berne et al., 2010*). Briefly, PicoGreen was diluted 1:200 in TE (10mM Tris–HCl, 1mM EDTA, pH 7.5) buffer prior analysis. A volume of 150 µl of planktonic phase cells grown in M2G was centrifuged (2 min at $15,000 \times g$, RT) to pellet cells and 100 µl of supernatant was added to 100 µl diluted PicoGreen reagent in a black 96-well plate with a clear flat bottom (Corning). After 5 min of incubation at room temperature, the fluorescence was measured using a Synergy HT microplate reader (using 485/528 Ex/Em nm filter set). A calibration curve (using Lambda DNA standard provided in the Quant-it kit, from 0 to 20 µg ml$^{-1}$) was performed for each measured plate. The fluorescence values measured for the standard samples were plotted versus the DNA concentration in those standards, and the linear regression from the standard curve was used to determine the DNA concentration from each sample.

### Biofilm quantification by crystal violet staining

*C. crescentus* cells were grown to mid-log phase in M2G medium under shaking conditions and diluted to $OD_{600} = 0.1$ in the same culture medium.

Aliquots of 1 ml of diluted cells were placed in the aforementioned growth conditions consisting of various aeration levels. After incubation at 30°C, the planktonic phase was carefully removed from the tube. The tube was rinsed with sterile dH$_2$O to remove non-attached cells, and the biomass attached to the inside of the tube was stained with a 0.1% crystal violet solution for 5 min and rinsed again with sterile dH$_2$O to remove excess crystal violet. The crystal violet stained cells were eluted with 10% acetic acid and quantified by measuring $A_{600}$.

### Growth curve measurements

Mid-log *C. crescentus* cultures grown in M2G medium under standard shaking conditions (3 ml cultures in 15-ml tubes, agitation at 300 rpm) were diluted to $OD_{600} = 0.15$ in the same culture medium and incubated at 30°C under moderate and maximal aeration conditions (see 'Bacterial strains and growth conditions' subsection). A volume of 2 µl was taken at different time points over the course of 48 hr from each culture, and $A_{600}$ was measured using a Nanodrop 2000c spectrophotometer (Thermo Scientific). Readings were corrected by a factor 10, to adjust for a 1-cm pathlength (and as determined by a standard curve using a WT culture [twofold dilutions in M2G] read once using the 2-µl drop option and a second time using the 1ml [in a 1-cm plastic cuvette] option of the Nanodrop spectrophotometer).

In addition, 150 µl were taken and frozen immediately at −20°C for eDNA quantification. All samples from one biological replicate (previously kept at −20°C) were assayed at once on the same plate, using the same calibration curve, as described above.

### Spent medium preparation

Spent medium was prepared as previously described (*Berne et al., 2010*). Briefly, bacteria were grown for 36 hr (late stationary phase) at 30°C in M2G medium. Cells were removed from the cultures by centrifugation (10 min at $8,000 \times g$, RT) and the supernatant (spent medium) was filter-sterilized using a 0.2-µm filter and kept at −20°C until used.

### Planktonic cells visualization by microscopy

Exponential phase cultures ($OD_{600} = 0.4–0.6$) were spotted on a glass coverslip (1 µl) and covered by a 1% agarose pad in M2G (Seakem LE agarose). Epifluorescence microscopy was performed as described below.

### Biofilm visualization by microscopy

For microscopy analysis, biofilms were grown in 2-ml microtubes under moderate aeration conditions as described above, with the only difference that a 22 × 5 mm strip of PVC coverslips (refs 12–547, Fisher Scientific) placed vertically in the microtube before sealing with Aeraseal. After incubation at

30°C, the coverslip was rinsed with sterile dH$_2$O to remove non-attached cells and stained for microscopy as described below (Live/Dead staining or FDG labeling).

## Live/Dead staining of cells inside the biofilm and quantification using microscopy images

Rinsed biofilm covered PVC strips (see above) were stained with the BacLight Bacterial Viability kit (L7007, Invitrogen). A volume of 0.5 µl of 1:1 Live/Dead stain mixture was added to 500 µl sterile dH$_2$O, placed on top of the strip and incubated for 15 min at room temperature in the dark. The stained biofilm was then rinsed and observed by epifluorescence microscopy. Live cells quantification was performed on microscopy images obtained above as described previously (*Berne et al., 2010*). Fluorescent signals coming from the green (live cells) and the red (dead cells) channels were quantified using the ImageJ analysis software (*Schneider et al., 2012*), as described below. For a given picture, the areas for the green and red signals were added and adjusted to 100% and percentage of each signal was then calculated. Results are given as the average from 10 random images for each biofilm sample, performed in independent triplicates.

## FDG staining

The activity of the *parDE$_4$* promoter in the biofilm was monitored using the plac290-P*parDE$_4$* construct and the fluorescein Di-β-D-Galactopyranoside (FDG), a fluorescent substrate for β-galactosidase (*Rotman et al., 1963*). Rinsed PVC strips (see above) covered with biofilms were stained with FDG (Invitrogen) using the following procedure. Firstly, strips were rinsed with sterile dH$_2$O. Secondly, 1 µl of FDG was added to 100 µl sterile dH$_2$O, placed on top of the strip and incubated for 15 min at room temperature in the dark. Finally, the stained biofilm was rinsed again and observed by epifluorescence microscopy. Fluorescence quantification was performed using the ImageJ analysis software (*Schneider et al., 2012*). Intensity is defined as the average signal value measured on the detected particle. Results are given as the average intensity from 10 random images for each biofilm sample, performed in independent triplicates.

## Competition experiments in flow cells

Biofilm were grown in flow cells using the previously described setup (*Berne et al., 2010*). Mid-log phase cultures of WT and Δ*parDE$_4$* containing the mini*Tn7::gfp* or mini*Tn7::dsred* insertion were diluted to an OD$_{600}$ of 0.025 in M2G and mixed to 1:1 ratio prior inoculation in the flow cell (200 µl). Initial attachment was performed in the absence of flow for 1 hr, followed by a constant flow of 3 ml hr$^{-1}$. Surface colonization of the glass surface covering the flow cells was monitored for 7 days at room temperature. Epifluorescence images were recorded at different time points, the amount of each population was quantified (as the fluorescence area for green and red signals), and the ratio of each population was calculated (as a percentage of the total population for the given time point). Each experimental condition was performed twice (swapping GFP and dsRed labeling for each replicate) in parallel triplicate chambers.

## Cell dispersal quantification

The amount of cells released from the biofilm grown in a flow cell (as described above) was determined by counting the number of colony-forming units (CFU) in the flow through. A sample of 1 ml was collected 5 cm downstream of the flow cells. CFU were determined by generating 1:5 serial dilution of the flow through in M2G. Twenty µl of each dilution were spotted on a dry PYE-agar plate, in triplicate, and incubated at 30°C for 2 days. Only spots of serial dilutions which contained 5–30 colonies were considered for analysis. Plates were imaged using a ChemiDoc imaging system (Biorad), using the 530 and 605 nm emission fluorescence filter reading parameters, to allow quantification of GFP and DsRed-labeled colonies in each spot.

## β-Galactosidase promoter activity assays

β-Galactosidase activity was quantified colorimetrically in 96-well plates as described previously (*Miller, 1972*; *Berne et al., 2018a*, *Berne and Brun, 2019*). Overnight cultures of strains bearing plac290lacZ grown in M2G were diluted to OD$_{600}$ of 0.05 and incubated at 30°C until an OD$_{600}$ of 0.4–0.6 was reached. A culture volume of 200 µl was mixed with 600 µl of Z buffer (60 mM Na$_2$HPO$_4$,

40 mM NaH$_2$PO$_4$, 10 mM KCl, 1 mM MgSO$_4$, 50 mM β-mercaptoethanol). Cells were then permeabilized using 50 µl of chloroform and 25 µl of 0.1% sodium dodecyl sulfate. Two hundred µl of the substrate o-nitrophenyl-β-D-galactoside (4 mg ml$^{-1}$) were added to the permeabilized cells. Upon development of a yellow color, the reaction was stopped by raising the pH to 11 with addition of 400 µl of 1 M Na$_2$CO$_3$. Absorbance at 420 nm ($A_{420}$) was determined and the Miller Units of β-galactosidase activity were calculated as $[(A_{420})(1000)]/[(OD_{600})(t)(v)]$ where $t$ is the time in minutes and $v$ is the volume of culture used in the assay in ml. The β-galactosidase activity of CB15 plac290 (empty vector control) was used as a blank sample reference.

## Quantitative PCR

WT and ΔparDE$_4$ strains carrying the PparDE$_4$-lacZ plasmid were grown overnight under the four aeration conditions described above, then diluted to OD$_{600}$ of 0.05 and incubated at 30°C under the same aeration until an OD$_{600}$ of 0.4–0.6 was reached. Cells were centrifuged at 4°C and pellets were stored at −80°C until further use. RNA samples were prepared using the Monarch Total RNA Miniprep kit (New England Biolabs Inc), followed by a DNAse treatment using the turbo DNA-free kit (Life Technologies), and storage at −80°C. One-step quantitative RT-PCR was used to determine gene expression levels using the Luna Universal One-Step RT-qPCR kit (New England Biolabs Inc) according to the manufacturer's recommendations using 4 ng of total RNA per 20 µl reaction. The reactions were performed on a QuantStudio 3 device (Applied Biosystems, Thermo Fisher Scientific). Gene expression ratios were calculated using rpoD (CCNA_03142) as the reference gene and the ccoN (CCNA_01467) as the target gene. Sequences of the primers used are provided in *Supplementary file 2*. Each gene expression value was obtained from two independent biological replicates, grown on separate days. RT-qPCR reactions were run in triplicate. Gene expression ratios were calculated using the Pfaffl method (*Pfaffl, 2001*), which takes into account primer efficiencies.

## Epifluorescence microscopy and image analysis

Epifluorescence microscopy was performed using an inverted Nikon Ti2 microscope with a Plan Apo ×60 objective, a GFP/DsRed filter cube, an Andor iXon3 DU885 EM CCD camera, and Nikon NIS Elements imaging software.

Image analysis was performed using ImageJ functions and plug-ins (*Schneider et al., 2012*). For Live/Dead staining quantification, red and green images were analyzed separately. Sixteen-bit images were thresholded using the B/W default setting and the total area of fluorescent signal was automatically determined using the Analyze Particle function, as the fraction area. For fluorescence intensity measurements (FDG labeling, GFP, mCherry, or DsRed signals), regions of interest (ROI) corresponding to single cells or clusters of cells (for biofilm images) were first determined using the phase images using the Analyze Particle function. ROI were then added to the ROI manager and transferred to fluorescence images (red and green separately). Finally, integrated fluorescence intensity was measured in each ROI. Background fluorescence was calculated as the average of 5 different ROI devoid of cells for each image and subtracted to the mean fluorescence intensity of each particle.

## Acknowledgements

The authors thank Aretha Fiebig and the Crosson laboratory for providing strains and engaging discussions, Marylise Duperthuy for the use of the QuantStudio 3 apparatus, and the members of the Brun laboratory for their constant feedback. This study was supported by grant R35GM122556 from the National Institutes of Health and by a Canada 150 Research Chair in Bacterial Cell Biology to YVB.

# Additional information

## Funding

| Funder | Grant reference number | Author |
| --- | --- | --- |
| Canada Research Chairs | | Yves V Brun |

| Funder | Grant reference number | Author |
|--------|------------------------|--------|
| National Institutes of Health | R35GM122556 | Yves V Brun |

The funders had no role in study design, data collection, and interpretation, or the decision to submit the work for publication.

### Author contributions

Cecile Berne, Conceptualization, Resources, Formal analysis, Validation, Investigation, Visualization, Methodology, Writing - original draft, Writing – review and editing; Sébastien Zappa, Formal analysis, Validation, Investigation, Methodology, Writing – review and editing; Yves V Brun, Conceptualization, Supervision, Funding acquisition, Methodology, Project administration, Writing – review and editing

### Author ORCIDs

Cecile Berne (iD) http://orcid.org/0000-0003-3731-9317
Sébastien Zappa (iD) http://orcid.org/0000-0003-3190-9199
Yves V Brun (iD) http://orcid.org/0000-0002-9289-1909

### Decision letter and Author response

Decision letter https://doi.org/10.7554/eLife.80808.sa1
Author response https://doi.org/10.7554/eLife.80808.sa2

## Additional files

### Supplementary files

- Supplementary file 1. Table of strains used in this study.
- Supplementary file 2. Table of primers used in this study.
- MDAR checklist

### Data availability

All data generated or analyzed during this study are available on Dryad.

The following dataset was generated:

| Author(s) | Year | Dataset title | Dataset URL | Database and Identifier |
|-----------|------|---------------|-------------|-------------------------|
| Berne C, Zappa S, Brun Y | 2022 | Data from: eDNA-stimulated cell dispersion from Caulobacter crescentus biofilms upon oxygen limitation is dependent on a toxin-antitoxin system | https://dx.doi.org/10.5061/dryad.4j0zpc8fc | Dryad Digital Repository, 10.5061/dryad.4j0zpc8fc |

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
