## [Editor Report]

In this work, the authors present compelling evidence that a toxin-antitoxin system contributes to biofilm dispersal under oxygen limited conditions. This work makes important contributions to two areas of microbial physiology; functional understanding of toxin-antitoxin systems, which have remained largely elusive, and mechanistic regulation or biofilm dispersal, is a critical, but less understood aspect of biofilm physiology.

---

## [Decision Letter]

**Decision letter after peer review:**

Thank you for submitting your article "eDNA-stimulated cell dispersion from *Caulobacter crescentus* biofilms upon oxygen limitation is dependent on a toxin- antitoxin system" for consideration by *eLife*. Your article has been reviewed by 2 peer reviewers, and the evaluation has been overseen by a Reviewing Editor and Naama Barkai as the Senior Editor. The reviewers have opted to remain anonymous.

Essential revisions:

1) The authors attempt to measure parDE expression in different areas of a biofilm with a promoter-lacZ reporter in which activity of the reporter is measured with a substrate that becomes fluorescent when cleaved by the lacZ gene product. This is a clever approach, but it lacks normalization by cell density, a typical feature of assessments of promoter-lacZ fusions. On first pass, the results presented are identical to what one would expect based on differences in cell density alone and I don't believe any conclusions can be drawn about expression of parDE4 across the biofilms grown in static cultures. Given that the authors have already built PparDE4-GFP fusions, it would be quite straightforward to repeat the experiment presented in Figure 8A by comparing PparDE4-GFP signal with a constitutively expressed red fluorescent protein for normalization of cell density. However, the use of stable reporter proteins poses a problem as these would accumulate more in non-growing or slow-growing cells and not necessarily read out cell density. The best solution here therefore would be to measure expression by directly measuring RNA levels (qRT-PCR or RNA-seq) not via stable protein reporters.

2) While the introduction presents background on TA types, and known TA involvement in biofilms, there is no discussion about other known roles of TA function. In particular, post-segregational killing seems like a relevant topic since this is presumably a similar mechanism as what is being observed here (in that case, TA transcription stops due to plasmid loss, while here TA transcription is reduced due to O2-dependent regulation). Several of the points made in the discussion about TA systems/PCD could also be made in the introduction to more clearly set the reader up with the necessary context to appreciate the impact/contribution of this work (e.g. some of the controversy in the PCD/TA literature, the inverse relationship between TA transcription and activity).

3) The choice of media in each experiment is not clear, but as the authors know, has important implications for biofilm formation in terms of baseline fraction of cells which elaborate holdfast and sensitivity to surface stimulation of holdfast synthesis. Furthermore, no rationale is given for media choices. The authors should be explicit in each experiment about the growth conditions including media, or at least state that all physiological experiments are were done in the same medium. The methods section ambiguously states that cells were cultures in either PYE or M2G.

*Reviewer #1 (Recommendations for the authors):*

Suggested experiments to further support the key conclusion that the ParE toxin is responsible for cell death:

– Make a catalytically inactive toxin variant to see whether cell death requires the toxin activity.

– Measure ParE activity directly e.g. assess whether cells have altered sensitivity to gyrase inhibiting drugs, image cells at higher resolution (possibly with DAPI staining) to determine whether changes to cell morphology is consistent with gyrase inhibition.

Suggested experiments to address whether transcriptional repression of ParDE4 leads to release of the toxin and thus cell death:

– Measure protein levels of toxin and antitoxin by western blot in O2-limited cells to see if there is a change in ratio of toxin and antitoxin

– Shut off parDE transcription either by placing the locus under control of a repressible promoter or using CRISPRi, then measuring growth (or evidence of ParE activity) after repression. In both cases, growth inhibition consistent with ParE activity should now be O2-independent.

*Reviewer #2 (Recommendations for the authors):*

Given the timing of biofilm formation, in the experiments involving growth in a flow cell, it seems the authors have used defined medium in which wild-type cells have a low propensity for holdfast synthesis and attachment. Though not essential for this publication, I wonder if parDE plays a role in PCD and dispersal when cells are grown in richer, more complex PYE media where attachment is more robust.

I have several questions/comments about the measurements and experimental conditions for the static growth experiments:

The authors claim to quantify "cell death and eDNA release in these biofilms" (line 148) but the measurements of live/dead cells and eDNA are from the planktonic phase not the surface attached cells. I suspect the phenomena that parDE4 contributes to cell death and eDNA release under limited oxygen conditions is not limited to surface attached biofilm conditions, thus I don't question the data, but rather how it is discussed. I recommend the authors focus on "culture conditions" rather than "biofilm condition" in the early part of the paper where they establish that parDE4 contributes to cell death under oxygen limitation. Certainly the culture conditions affect biofilm dynamics, but I suspect these phenotypes do not require the cells to be in a biofilm. If the authors wanted to formally examine this idea, and specifically test if the cells needed to be in a biofilm to 'feel' the effects of parDE4 and low oxygen, they could repeat the static culture experiments in a holdfast null strain. I predict that cell death and eDNA release would be comparable to holdfast competent cells, without the potentially confounding effects of cell attachment.

Figure S1A: Please be more explicit about how the growth measurement were made. Did you measure the same tube over time? Were the cultures mixed prior to measuring OD? If so, how did you maintain the aeration condition? Or did you set up parallel tubes and make terminal measurements at each time point? Growth medium?

Line 36: use of an acronym in abstract without definition

Line 79-80: perhaps emphasize "new-born swarmer" and consider "dissociate" instead of "disperse".

Line 150: 'viability' was not measured. Please rephrase to reflect the experimental measurement.

Please add scale bars to all microscopy images (Figure 2 and 4).

line 189: Similar to other comments above, the authors should rephrase point 1 as they don't measure cell death "in the biofilm". Perhaps instead: "ParD4 has a protective effect against cell lysis that enhances biofilm formation".

The red signal corresponding to dead cells is very difficult to see in Figure 2. From the images as presented, it doesn't seem that 40% of WT cells are dead. Consider showing the channels separately and merged. Also given the difficulty if seeing sparse signal on a black background, consider scaling the images differently (for example from 0=white to max=color, or present the single channels as inverted BW images).

Line 181: activation of TAS is not just about "production" of the proteins. More generally activation comes when there is an imbalance in steady state levels, and reduced stability of antitoxins compared to toxins is an important contributor to these imbalances. The discussion should be enhanced.

The authors should clarify metrics used to quantify microscopy images. It seems that the authors use thresholded areas in a binary fashion that ignores information from signal intensity. But it in these multidimensional structures, areas with thicker cell masses should have higher signal than areas with monolayers of cells. By simply thresholding the images and quantifying area, the authors loose a dimension of the data. Summing intensity in areas where signals overlap seems more informative than simply treating the pixels as binary.

Figure 6D, would be better presented more parallel to 6C such that the Y-axis reflected "relative biofilm formation (∆parDE4 / WT)" rather than "biofilm formation (% of WT)". This would also be more consistent with the language in line 280.

It is not clear why some experiments assess ccoN expression via a promoter reporter and others via transcript abundance by RT-PCR. Nevertheless, the RT-PCR experiments should be labeled as ccoN transcript levels rather than PccoN expression (line 273 and Figure 6B). Also I suspect that ccoN transcript is essentially undetectable under maximal aeration conditions leading to a denominator of practically 0 in the "fold-change". While I have no question about the veracity of the data, I prefer non-normalized (non-ratioed) data whenever possible, especially when one condition is essentially "off". Just my two cents.

Line 306-7: wording is somewhat confusing.

Figure 7 and S3: I have several suggestions for data presentation. Make the data points smaller in the scatter plots to better visualize the density of points near the axis. Consider plotting on a log scale rather than a linear scale to better spread the points with lower intensity and highlight the bulk of the cells rather than the handful of outliers which are what is primarily obvious on the linear scaling.

Please indicate how the threshold was determined to assess if each promoter was on or off for figure 7A. How many cells have "both"? Is "non-labeled" the best description of cells in which neither promoter was sufficiently active to be called "on"? Even "non-fluorescent" seems better than "non-labeled" as label implies staining.

Line 505: the authors report using 10 ug/ml of chloramphenicol to select for plasmids in Caulobacter. This seems extremely high and quite inhibitory even in the presence of a Cm resistance gene. Typically for Caulobacter 1-2 ug/ml chlor is sufficient to maintain a plasmid. Please check if this number is correct.

---

## [Author Response]

Essential revisions:1) The authors attempt to measure parDE expression in different areas of a biofilm with a promoter-lacZ reporter in which activity of the reporter is measured with a substrate that becomes fluorescent when cleaved by the lacZ gene product. This is a clever approach, but it lacks normalization by cell density, a typical feature of assessments of promoter-lacZ fusions. On first pass, the results presented are identical to what one would expect based on differences in cell density alone and I don't believe any conclusions can be drawn about expression of parDE4 across the biofilms grown in static cultures. Given that the authors have already built PparDE4-GFP fusions, it would be quite straightforward to repeat the experiment presented in Figure 8A by comparing PparDE4-GFP signal with a constitutively expressed red fluorescent protein for normalization of cell density. However, the use of stable reporter proteins poses a problem as these would accumulate more in non-growing or slow-growing cells and not necessarily read out cell density. The best solution here therefore would be to measure expression by directly measuring RNA levels (qRT-PCR or RNA-seq) not via stable protein reporters.

We agree with that the experiment presented in Figure 8A was lacking some internal controls. While we agree that qPCR or RNA-seq would be good options to assess *parDE_4_* expression in the different portions of the biofilm (top, middle and bottom layers), technical reasons precluded this. First, our experimental set-up (about 1/3 of a 1.8 x 0.5 cm piece of plastic coverslip) didn't provide sufficient material for RNA extraction for suitable qPCR or RNA-seq experiments. Furthermore, both the toxin and antitoxin genes are short (303 and 291 bp respectively), providing limited options for qPCR primers. We tested many primer pairs but in each case, the denaturation curve showed a lot of non-specific amplification.

As an alternative approach, and as suggested by the reviewer, we constructed a strain constitutively expressing the red fluorescence protein Dsred (similarly to the one we use for the competition experiments in Figure 4) and harboring the P*parDE_4_*-*lacZ* construct we are using throughout this work (Figures 3C and 6B). We repeated the experiments and monitored *parDE_4_* expression in different biofilm layers using our promoter-*lacZ* reporter and FDG (a fluorogenic dye cleaved by ß-galactosidase to release fluorescein) and the red fluorescence signal produced by Dsred. The red fluorescence produced by the constitutively expressed Dsred was used as a measure of cell density to normalize the results. These new results now replace the previous data in Figure 8A and lead to the same conclusions as in the initial submission. In addition, we repeated these experiments using a strain harboring the same constitutively expressed Dsred protein and the O_2_ regulated P*ccoN*-*lacZ* construct we used in Figure 3C to provide a measure of O_2_ availability in the two lower biofilm areas.

2) While the introduction presents background on TA types, and known TA involvement in biofilms, there is no discussion about other known roles of TA function. In particular, post-segregational killing seems like a relevant topic since this is presumably a similar mechanism as what is being observed here (in that case, TA transcription stops due to plasmid loss, while here TA transcription is reduced due to O2-dependent regulation). Several of the points made in the discussion about TA systems/PCD could also be made in the introduction to more clearly set the reader up with the necessary context to appreciate the impact/contribution of this work (e.g. some of the controversy in the PCD/TA literature, the inverse relationship between TA transcription and activity).

We agree and we now discuss this in both the introduction and Discussion sections in this new version.

Lines 103-130 (introduction): " TAS are widespread in bacterial and archaeal genomes, but despite their abundance, the biological relevance of most TAS is still elusive (Fraikin et al., 2020). TAS were first described as plasmid addiction modules that ensure plasmid stabilization via post-segregational killing of plasmid-free cells (Ogura and Hiraga, 1983, Gerdes et al., 1986). TAS have also been shown to promote addiction to certain chromosomally-encoded elements such as integrative conjugative elements (Wozniak and Waldor, 2009) or CRISPR-cas loci (Li et al., 2021). In addition, TAS have been described as defense mechanism against phage infection where host translation is inhibited by the phage (Pecota and Wood, 1996, Fineran et al., 2009, Song and Wood, 2020, LeRoux and Laub, 2022, Vassallo et al., 2022). In bacterial cells that lost their plasmid / chromosomal element encoding the TAS, or that are infected by phage, the amount of labile antitoxin rapidly decreases, leading to toxin activation and subsequent cell death. In addition to the well accepted role of TAS in addiction and phage exclusion, TAS have been linked to diverse physiological responses, such as biofilm formation, stress response and persistence (Kamruzzaman et al., 2021), although this is still debated (Ronneau and Helaine, 2019, Wade and Laub, 2019, Song and Wood, 2020, Jurėnas et al., 2022). Many TAS have been reported to be transcriptionally upregulated under environmental stress conditions (Jurėnas et al., 2022), but this increase does not necessarily trigger liberation of an active toxin (LeRoux et al., 2020). "

Lines 519-544 (discussion): " The system described in this study is conceptually similar to TAS involved in plasmid maintenance / postsegregational killing or phage defence mechanisms. Indeed, TAS encoded on a plasmid, such as the CcdAB (Ogura and Hiraga, 1983) or Hok/Sok (Gerdes et al., 1985) pairs in *E. coli*, ensure plasmid maintenance over generations: if a daughter cell fails to inherit the plasmid bearing the TAS upon division, the amount of labile antitoxins in this cell quickly drops, allowing free toxins to kill. A similar strategy is seen in instances of phage neutralization upon infection (LeRoux and Laub, 2022) by TAS such as ToxIN (Fineran et al., 2009) and DarTG (LeRoux et al., 2022) for example. Phage infection usually provokes inhibition of the host cell transcription, followed by degradation of the antitoxins and liberation of the toxins. Infected cells are rapidly killed, preventing massive phage replication. In this example, killing a few cells via PCD neutralizes phage infection and rescues the rest of the colony. We propose the ParDE_4_-mediated PCD we described in this study might fulfill a similar role: selectively killing a few cells in a crowded biofilm where conditions are deleterious, to promote the dispersal of newborn cells from the deleterious conditions."

3) The choice of media in each experiment is not clear, but as the authors know, has important implications for biofilm formation in terms of baseline fraction of cells which elaborate holdfast and sensitivity to surface stimulation of holdfast synthesis. Furthermore, no rationale is given for media choices. The authors should be explicit in each experiment about the growth conditions including media, or at least state that all physiological experiments are were done in the same medium. The methods section ambiguously states that cells were cultures in either PYE or M2G.

We apologize for the confusion. All experiments have been done in M2G. PYE was only used during strain construction and cloning procedures. We now provide this information in the "Material and Method section" (Lines 619-620: "All experiments were carried out using M2G, while *C. crescentus* was grown in PYE for strain construction."). M2G usage was already mentioned in the relevant subsection of the Material and Method section, and we now also added this information in figure legends when appropriate.

We used M2G in all our experiments for several reasons: Experiments in our previous related work on cell death in the biofilm (Berne et al. (2010) Mol Microbiol) were done in M2G. M2G is also the reference medium we used in our earlier works on holdfast regulation and biofilm formation (Berne et al. (2018) Mol Microbiol; Berne et al. (2019) J Bact).

Reviewer #1 (Recommendations for the authors):Suggested experiments to further support the key conclusion that the ParE toxin is responsible for cell death:– Make a catalytically inactive toxin variant to see whether cell death requires the toxin activity.– Measure ParE activity directly e.g. assess whether cells have altered sensitivity to gyrase inhibiting drugs, image cells at higher resolution (possibly with DAPI staining) to determine whether changes to cell morphology is consistent with gyrase inhibition.

We thank the reviewer for these suggestions. While the present work focusses on the link between O_2_ deprivation, *parDE_4_* expression, and biofilm control, our next step will be to dissect the mechanism of action of the ParD_4_-ParE_4_ pair in this context. We did assess cell morphology as suggested and can already say that we do not observe noticeable changes in cell morphology under conditions where ParDE_4_-correlated cell death is observed (O_2_-limited conditions). We have added these observations as Figure 7—figure supplement 2, and in the Discussion section (Lines 544-546: "Under conditions where ParDE_4_-correlated cell death is observed (O_2_-limited conditions), cell morphology does not change significantly (Figure 8 and Figure 7—figure supplement 2)").

Suggested experiments to address whether transcriptional repression of ParDE4 leads to release of the toxin and thus cell death:– Measure protein levels of toxin and antitoxin by western blot in O2-limited cells to see if there is a change in ratio of toxin and antitoxin– Shut off parDE transcription either by placing the locus under control of a repressible promoter or using CRISPRi, then measuring growth (or evidence of ParE activity) after repression. In both cases, growth inhibition consistent with ParE activity should now be O2-independent.

Here again, we think these are very interesting suggestions and will be considered in our future experiments, as the presented work here does not focus on the mode of action of the ParDE_4_ TAS, but rather its relationship with biofilm formation and cell dispersal under deleterious conditions (aka O_2_ deprivation).

In addition, we previously tried to obtain antibodies for both ParD_4_ and ParE_4_ without success. First of all, we had antibodies raised against the entire ParD_4_ purified protein (in rabbit) produced and purified, but, despite our numerous efforts, we failed to detect a band in Western blots using them, even when using the overexpressing strain carrying the plasmid pMT686-*parD_4_* that clearly induces the expression of ParD_4_ in our hands here and in previous work published in the Crosson laboratory (Fiebig *et al.* (2010) Mol. Microbiol.). Second, the company we used for antibody production failed to overexpress ParE_4_, despite multiple tries over several months. We are currently exploring other routes for antibody production.

We also tried several strategies to overexpress ParE_4_ and failed to do so in both *E. coli* and *Caulobacter*. This protein seems highly toxic, even for *E. coli*, and none of our efforts have been successful. We are currently designing new strategies and this will be part of our future exploration of ParDE_4_ mode of action. This aspect is now included in the Discussion section (Lines 553-557: "So far, we have failed to express the parE_4_ gene both in *E. coli* and C. crescentus, suggesting that its product is highly toxic. Our future efforts will aim at engineering an attenuated version of parE_4_. Such a tool will help to identify its targets and elucidate the pathway that leads to cell killing by this TAS.").

Reviewer #2 (Recommendations for the authors):Given the timing of biofilm formation, in the experiments involving growth in a flow cell, it seems the authors have used defined medium in which wild-type cells have a low propensity for holdfast synthesis and attachment. Though not essential for this publication, I wonder if parDE plays a role in PCD and dispersal when cells are grown in richer, more complex PYE media where attachment is more robust.

We address our choice of using defined M2G medium in the reply to essential revisions above. Nevertheless, we tested biofilm formation of the different single ParDE and RelBE mutants used in this study in PYE and obtained similar results. The ∆*parDE_4_* mutant is the only tested mutant to behave differently than WT in PYE, and, under moderate aeration, we can measure an about 15% increase of biofilm (compared to about 30% in M2G). This attenuated response is not that surprising, as previous work by us and other laboratories report a higher dynamic range of holdfast related phenotypes in defined minimal M2 media (M2G or M2X, see Fiebig et al. 2014, Berne et al. 2018, Berne and Brun 2019). We are reluctant to add these data to the present manuscript, as all the presented experiments were done in M2G and we think it could add confusion.

**Author response image 1. sa2fig1:** Biofilm formation, quantified by crystal violet staining; results are expressed as a percentage of biofilm formed compared to WT. Results are given as the average of two-four independent experiments, each run in duplicate, and the error bars represent the Standard Error of the Mean (SEM). Statistical comparisons are calculated using Student’s unpaired t-tests; only samples statistically different from WT are shown. * *P* < 0.05; ** *P* < 0.01.

I have several questions/comments about the measurements and experimental conditions for the static growth experiments:The authors claim to quantify "cell death and eDNA release in these biofilms" (line 148) but the measurements of live/dead cells and eDNA are from the planktonic phase not the surface attached cells. I suspect the phenomena that parDE4 contributes to cell death and eDNA release under limited oxygen conditions is not limited to surface attached biofilm conditions, thus I don't question the data, but rather how it is discussed. I recommend the authors focus on "culture conditions" rather than "biofilm condition" in the early part of the paper where they establish that parDE4 contributes to cell death under oxygen limitation. Certainly the culture conditions affect biofilm dynamics, but I suspect these phenotypes do not require the cells to be in a biofilm. If the authors wanted to formally examine this idea, and specifically test if the cells needed to be in a biofilm to 'feel' the effects of parDE4 and low oxygen, they could repeat the static culture experiments in a holdfast null strain. I predict that cell death and eDNA release would be comparable to holdfast competent cells, without the potentially confounding effects of cell attachment.

We agree with the reviewer that section was misleading and we and modified the text to be clearer regarding our conclusions. We first changed to "We tested the ability of the TAS mutants to form biofilms after 48 hours, and quantified cell death and eDNA release under these growth conditions"(line 179, instead of "…cell death and eDNA release in these biofilms"), and then rewrote the conclusion sentence to " These results suggest that ParDE_4_ plays a role in cell death and eDNA release under our experimental conditions, and that the observed changes in eDNA concentration yield differences in biofilm regulation. " (lines 187-189, instead of "… a role in biofilm regulation and eDNA release under our experimental conditions"). We also modified the text in the end of section 2, to " Furthermore, these results indicate that ParDE_4_ is involved in stimulating cell death and eDNA release, yielding a change in biofilm formation." (lines 220-226, instead of "… cell death and eDNA release in biofilms").

As suggested by the reviewer, we added a new supplementary figure (Figure 1—figure supplement 2) providing eDNA release and percentage of dead cells in the planktonic phase in a holdfast parDE_4_ double mutant (∆parDE_4_ ∆hfsDAB). As expected, this double mutant exhibits similar phenotypes compared to the parDE_4_ single mutant (lower cell death and subsequent eDNA release compared to WT), showing that, indeed, cell death and eDNA release respond to oxygen availability and do not require cells to be able to adhere and form a biofilm. We added these observations in the result section "To test if this phenotype was specific for *C. crescentus* cells that are able to form biofilms, we knocked out the hfsDAB holdfast synthesis cluster in the ∆parDE_4_ background to generate a strain unable to produce holdfast, and therefore unable to adhere to surfaces. The double mutant ∆parDE_4_ ∆hfsDAB phenocopied the ∆parDE_4_ strain, with lower eDNA and lower proportion of dead cells (Figure 1—figure supplement 2). These results indicate that the function of ParDE_4_ does not require cells to be adhered to a surface and suggest that it might be responding to the differences in medium aeration as described in a later section." (lines 189-204).

Figure S1A: Please be more explicit about how the growth measurement were made. Did you measure the same tube over time? Were the cultures mixed prior to measuring OD? If so, how did you maintain the aeration condition? Or did you set up parallel tubes and make terminal measurements at each time point? Growth medium?

We now provide all this information in the new "Growth curve measurements." subsection of "Material and method": " Growth curve measurements. Mid-log *C. crescentus* cultures grown in M2G medium under standard shaking conditions (3 ml cultures in 15 ml tubes, agitation at 300 rpm) were diluted to OD_600_ = 0.15 in the same culture medium and incubated at 30°C under moderate and maximal aeration conditions (see "Bacterial strains and growth conditions" subsection above). A volume of two µl was taken at different time points over the course of 48 h from each culture, and A_600_ was measured using a Nanodrop 2000c spectrophotometer (Thermo Scientific). Readings were corrected by a factor 10, to adjust for a 1 cm pathlength (and as determined by a standard curve using a WT culture (2-fold dilutions in M2G) read once using the 2 µl drop option and a second time using the 1 ml (in a 1 cm plastic cuvette) option of the Nanodrop spectrophotometer)." (lines 691-708).

Line 36: use of an acronym in abstract without definition

Corrected.

Line 79-80: perhaps emphasize "new-born swarmer" and consider "dissociate" instead of "disperse".

Corrected.

Line 150: 'viability' was not measured. Please rephrase to reflect the experimental measurement.

Corrected to "The percentage of dead cells was lower". (line 181)

Please add scale bars to all microscopy images (Figure 2 and 4).

Done.

line 189: Similar to other comments above, the authors should rephrase point 1 as they don't measure cell death "in the biofilm". Perhaps instead: "ParD4 has a protective effect against cell lysis that enhances biofilm formation".

Corrected.

The red signal corresponding to dead cells is very difficult to see in Figure 2. From the images as presented, it doesn't seem that 40% of WT cells are dead. Consider showing the channels separately and merged. Also given the difficulty if seeing sparse signal on a black background, consider scaling the images differently (for example from 0=white to max=color, or present the single channels as inverted BW images).

We now added a supplementary figure (Figure 2—figure supplement 1), to show the separated green and red channels, as well as inverted signals converted in BW, for a better visualization.

Line 181: activation of TAS is not just about "production" of the proteins. More generally activation comes when there is an imbalance in steady state levels, and reduced stability of antitoxins compared to toxins is an important contributor to these imbalances. The discussion should be enhanced.

Corrected to "In TAS, cell death usually occurs when there is an imbalance in steady state levels of toxins and antitoxins produced in the cell". (lines 231-232)

The authors should clarify metrics used to quantify microscopy images. It seems that the authors use thresholded areas in a binary fashion that ignores information from signal intensity. But it in these multidimensional structures, areas with thicker cell masses should have higher signal than areas with monolayers of cells. By simply thresholding the images and quantifying area, the authors loose a dimension of the data. Summing intensity in areas where signals overlap seems more informative than simply treating the pixels as binary.

We now provide all the relevant information in the new "Epifluorescence microscopy and image analysis" subsection in the Material and Methods: " … Image analysis was performed using ImageJ functions and plug-ins (Schneider et al., 2012). For Live / Dead staining quantification, red and green images were analyzed separately. 16-bit images were thresholded using the B/W default setting and the total area of fluorescent signal was automatically determined using the Analyze Particle function, as the fraction area. For fluorescence intensity measurements (FDG labelling, GFP, mCherry, or DsRed signals), regions of interest (ROI) corresponding to single cells or clusters of cells (for biofilm images) were first determined using the phase images using the Analyze Particle function. ROI were then added to the ROI manager and transferred to fluorescence images (red and green separately). Finally, integrated fluorescence intensity was measured in each ROI. Background fluorescence was calculated as the average of 5 different ROI devoid of cells for each image and subtracted to the mean fluorescence intensity of each particle". (lines 830-846).

We used thresholded images when quantifying Live/Dead staining images: for those experiments we needed an overall response (what is the area representing the red or green signal over the total field of view?), and each channel (red or green) was treated independently. For experiments where we measured fluorescence intensity, we indeed measured the integrated fluorescence intensity signal. We added this information in the method section ("Epifluorescence microscopy and image analysis").

Figure 6D, would be better presented more parallel to 6C such that the Y-axis reflected "relative biofilm formation (∆parDE4 / WT)" rather than "biofilm formation (% of WT)". This would also be more consistent with the language in line 280.

Corrected. Figure 6D has been modified to represent "ratio biofilm formation (∆*parDE_4_* / WT).

It is not clear why some experiments assess ccoN expression via a promoter reporter and others via transcript abundance by RT-PCR. Nevertheless, the RT-PCR experiments should be labeled as ccoN transcript levels rather than PccoN expression (line 273 and Figure 6B).

In Figure 5, we assayed both *ccoN* and *parDE_4_* promoter activities using promoter-*lacZ* fusions to assay both promoter using the same method. As mentioned above in the "Essential revisions" section, our several attempts to measure *parDE_4_* expression by qPCR were unsuccessful despite our best efforts. The *lacZ* fusions provided a good option, especially as we are using them in Figure 8 to measure P*parDE_4_* expression in biofilms by microscopy. In addition, we now include the P*ccoN-lacZ* fusion to measure *ccoN* expression in biofilms (in Figure 8).

In Figure 6, we wanted to assay *ccoN* and *parDE_4_* expressions in the same cultures. *parDE_4_* expression was assayed using *lacZ* fusion (our only current option, see above), and, as we couldn't use at the same time the P*ccoN-lacZ* construct to quantify *ccoN* expression in the same tube, we used qPCR instead. We realize that this information was not clearly provided in the previous version of the manuscript and we have corrected it here (in the result and method sections, as well as in Figure 6 legend).

When *ccoN* expression was assayed using qPCR, we now corrected to "*ccoN* transcript levels" (Figure 6 and in the text).

Also I suspect that ccoN transcript is essentially undetectable under maximal aeration conditions leading to a denominator of practically 0 in the "fold-change". While I have no question about the veracity of the data, I prefer non-normalized (non-ratioed) data whenever possible, especially when one condition is essentially "off". Just my two cents.

We agree that representing of qPCR data can indeed be tricky for the reasons you explained. We chose to represent the data as classical qPCR ratios using a reference condition (Maximal aeration conditions). This choice was driven by the fact that we wanted to represent these data along the P*parDE_4_-lacZ* data, and we wanted to have a value for the "maximal aeration" condition (reference condition, set to 1 as base ratio). We now added this information in the figure legend ("*qPCR (*calculated as described in the Material and Method section) were normalized to the on in "Maximal aeration" conditions set to 1").

This reference condition, while consisting in low transcription of *ccoN*, is still significant. The Ct of *ccoN* transcript is approximately 32 and reproducible for the reference condition and reaches 18-21 in the other tested conditions. The reason why we feel that the representation as ratios is valid is because our gene of interest (*ccoN*) and housekeeping gene (*rpoD*) are amplified with primers with rather important differences regarding their respective PCR efficiencies (≈80% and ≈96% respectively). As a result, the use of the Pfafl method and ratio representation (as mentioned in the method section) was appropriate for data analysis.

Line306-7: wording is somewhat confusing.

The sentence was deleted (it was redundant with the next one).

Figure 7 and S3: I have several suggestions for data presentation. Make the data points smaller in the scatter plots to better visualize the density of points near the axis. Consider plotting on a log scale rather than a linear scale to better spread the points with lower intensity and highlight the bulk of the cells rather than the handful of outliers which are what is primarily obvious on the linear scaling.

We modified these figures. The data points are now smaller for a better readability. While still being linear scale, we are now including, in Figure 7—figure supplement 1, representations using one-segment axes (like in the original submission), and two-segment axes for a better visualization of lower intensity datapoints. Anti-correlation curves are usually reported using linear scale graphs and we decided to keep linear scales axes. We felt that the anti-correlation of our data was better reflected using linear scale data compared to log scale ones, as illustrated in author response image 2.

**Author response image 2. sa2fig2:** Red and green fluorescence intensity of single cells grown under maximal, high, moderate, and limited aeration (all conditions combined). The same data are represented three different ways. data reported using single linear axes (Figure 7 and Figure 7—figure supplement 1). data reported using two-segment linear axes (Figure 7—figure supplement 1). Data represented using log10 axes.

Please indicate how the threshold was determined to assess if each promoter was on or off for figure 7A. How many cells have "both"? Is "non-labeled" the best description of cells in which neither promoter was sufficiently active to be called "on"? Even "non-fluorescent" seems better than "non-labeled" as label implies staining.

The images were not thresholded prior to quantification in this experiment. We now provide all the information in the method section ("Epifluorescence microscopy and image analysis" subsection), as explained above. A cell was considered expressing GFP or mCherry if the fluorescence signal was at least 1.2 times the background (information added in Figure 7 figure legend). If a cell expressed both GFP and mCherry signals, it was included twice in the quantification (once for each channel). Interestingly, a very small percentage cells carried both green and red fluorescence, as presented in Figure 6B and in the result section.

We agree with the reviewer that "non-labeled" was not the most appropriate word to use, and we changed the legend from "non-labelled" to "non-fluorescent" cells, as suggested.

Line 505: the authors report using 10 ug/ml of chloramphenicol to select for plasmids in Caulobacter. This seems extremely high and quite inhibitory even in the presence of a Cm resistance gene. Typically for Caulobacter 1-2 ug/ml chlor is sufficient to maintain a plasmid. Please check if this number is correct.

This was indeed a mistake and the correct Cm concentration is 1 µg/ml. It is now correct in the text.